# ZIN: When and How to Learn Invariance Without Environment Partition?

**Yong Lin**
HKUST
ylindf@connect.ust.hk

**Shengyu Zhu**[*]
Huawei Noah's Ark Lab
zhushyu@outlook.com

**Lu Tan**
Tsinghua University
tanl21@mails.tsinghua.edu.cn

**Peng Cui**
Tsinghua University
cuip@tsinghua.edu.cn

## Abstract

It is commonplace to encounter heterogeneous data, of which some aspects of the data distribution may vary but the underlying causal mechanisms remain constant. When data are divided into distinct environments according to the heterogeneity, recent invariant learning methods have proposed to learn robust and invariant models using this environment partition. It is hence tempting to utilize the inherent heterogeneity even when environment partition is not provided. Unfortunately, in this work, we show that learning invariant features under this circumstance is fundamentally impossible without further inductive biases or additional information. Then, we propose a framework to jointly learn environment partition and invariant representation, assisted by additional auxiliary information. We derive sufficient and necessary conditions for our framework to provably identify invariant features under a fairly general setting. Experimental results on both synthetic and real world datasets validate our analysis and demonstrate an improved performance of the proposed framework. Our findings also raise the need of making the role of inductive biases more explicit when learning invariant models without environment partition in future works. Codes are available at https://github.com/linyongver/ZIN_official.

## 1 Introduction

Machine learning algorithms with empirical risk minimization (ERM) generally assume independent and identically distributed (i.i.d.) data in training and test sets. Due to changing circumstances, selection bias, or time shifts, data distributions are often heterogeneous across different environments in practical applications. When the distributions of training and test data are indeed different, model performance can severely degrade [47, 17, 38], as ERM based algorithms may exploit the spurious correlations that are particularly useful to ERM in the training data but do not exist in the test data.

To tackle the distribution-shift or out-of-distribution generalization problem, a recent line of methods propose to utilize the causally invariant mechanisms (rather than the spurious correlations in the training data) that are supposed to be stable across different environments. For example, Peters et al. [36] proposes to exploit the invariance principle to learn a linear model that merely relies on the direct causes of the target. Such a model is shown to be robust to potential interventions on any variable

---

[*]Corresponding author.

36th Conference on Neural Information Processing Systems (NeurIPS 2022).

except the target itself. Arjovsky et al. [4] propose invariant risk minimization (IRM) to capture invariant correlations by learning representations that elicit an optimal invariant predictor across multiple training environments. Ahuja et al. [2], Jin et al. [21], Krueger et al. [24], Xie et al. [52], Chen et al. [8, 9] further develop variants of IRM by introducing game theory, regret minimization, variance penalization, multi-objective optimization, etc., and Xu and Jaakkola [54], Chang et al. [7], Lin et al. [26] try to learn invariant features by coupled adversarial neural networks. Some recent works report that IRM is less effective when applied to large neural networks [26, 15]. Lin et al. [27] find that this can be largely attributed to the overfitting problem, and Zhou et al. [57, 56] propose to alleviate this issue through imposing sparsity constraint and sample reweighting, respectively.

Noticeably, the aforementioned invariant learning methods require datasets to be explicitly partitioned into environments (or domains) according to the heterogeneity underlying the data. However, such an environment partition may be unavailable or hard to obtain in practice [29, 10]. Recently, a line of works try to learn invariance without environment indexes where the dataset is assembled by merging data from multiple environments. For instance, Creager et al. [10] propose environmental inference for invariant learning (EIIL), a two-stage method by firstly inferring the environments with an ERM based trained biased model and then performing invariant learning on inferred environments. Liu et al. [29] devise an interactive mechanism, called heterogeneous risk minimization (HRM), with environment inference and invariant learning on raw feature level, and Liu et al. [30] extend it to representation level with kernelized trick.

However, as we will show, learning invariant models under this circumstance (with only input-label pairs) can be *risky*, both theoretically and empirically. As our first contribution, in Section 4, we prove that learning invariant features from heterogeneous data without environment indexes is in theory impossible. Specifically, we provide a counter-example and a general theoretic result (Theorem 1), to show that the causally invariant features are unidentifiable if no environment information is provided. This impossibility result, similar to the identifiability issue in causal discovery [45, 37], independent component analysis (ICA) [20], and unsupervised learning of disentangled representations [32], motivates us to consider when and how to learn invariant features.

In this work, we turn to additional auxiliary variables that encode some information about the latent heterogeneity. This strategy can be treated as a case between the two existing directions: 1) it is less restrictive than requiring ideal environment partition according to distribution heterogeneity; and 2) it also obtains theoretic guarantee which is missing in the case with no environment information at all. Notably, such auxiliary information is often cheaply available for every input in practice [53, 51]. Examples include time index of the data in time series forecasting tasks [6, 33], locations (longitude and latitude) of collected satellite data in remote sensing [14, 40], and meta information associated with an instance [31]. Based on the additionally observed variables, we proceed to propose a framework to jointly learn environment partition and invariant representation in Section 5. Under a fairly general setting, we derive both sufficient and necessary conditions for our framework to identify invariant features. Extensions to other settings are discussed in Appendix A.

Finally, Section 6 presents experimental analysis on both synthetic and real world datasets. The proposed framework achieves an improved performance over existing methods like EIIL and HRM, and has a comparable performance to IRM with ground-truth environment partition.

## 2   Related Work

Invariant learning methods can be interpreted as robust to certain interventions or heterogeneity from a causal perspective [36]. On the other hand, heterogeneous data have also placed challenges and potential benefits in causal inference tasks. For example, Huang et al. [19] consider heterogeneous datasets to identify variables with changing local mechanism and further the causal directions. Here confounders are assumed as functions of the domain index or time. Tillman and Spirtes [48], Danks et al. [11], Huang et al. [18] study the multiple dataset setting with non-identical sets of variables for causal discovery, where the datasets may have different exogenous noise distributions and only part of the variables is present in each dataset.

Besides the invariant learning methods described in Section 1, another large class of methods for generalizing beyond training data is distributionally robust optimization (DRO) [5, 43, 25, 13, 12, 55]. DRO methods propose to optimize the worst-case risk over a set of distributions close to the training distribution. A notable instance of DRO methods is group DRO, which optimizes the worst-case loss over groups in the training data [41]. As with most invariant learning methods, group DRO generally requires a partition of groups obtained according to group annotation and label of each data sample.

Obtaining group annotations, however, can be costly or even impossible in practice. We may not know *a priori* the inherent spurious features associated with data. Thus, a number of works have proposed to infer group partition or directly identify the minority group by looking at features produced by biased models, e.g., [44, 1, 10, 28]. Notice that sometimes a small number of labelled annotations are still needed to find a proper trained model, which is different from the setting considered in this work. For example, Liu et al. [28] upweights samples with high losses from the initial ERM model and relies on a small validation set of annotated data to tune parameters. Additionally, Nam et al. [34], Sanh et al. [42] try to obtain a more robust model by boosting from the wrongly specified samples, based on the observation that models with limited capacity tend to learn spurious correlations or shortcuts. Unfortunately, as we will show in Section 4, it is in theory impossible to distinguish spurious or invariant features from only the observed data consisting of input-label pairs.

## 3 Preliminaries

Throughout this paper, upper-cased letters such as $\boldsymbol{X} \in \mathbb{R}^d$ and $Y \in \mathbb{R}$ denote random variables, and lower-cased letters such as $\boldsymbol{x}$ and $y$ denote deterministic instances. Suppose there exist invariant features $\boldsymbol{X}_v \in \mathbb{R}^{d_v}$ and spurious (non-invariant) features $\boldsymbol{X}_s \in \mathbb{R}^{d_s}$. We observe a scrambled version $\boldsymbol{X} = q(\boldsymbol{X}_v, \boldsymbol{X}_s) \in \mathbb{R}^d$ with $q$ being an injective function. Consider the dataset $\mathcal{D} := \{(\boldsymbol{x}_i, y_i)\}_{i=1}^n$ where data are collected from multiple environments $e \in \mathcal{E}_{supp}$. We use superscript $e$ to indicate the environment index with a variable, e.g., $\boldsymbol{X}^e$ and $Y^e$, when it is useful to make the index explicit. Notice that the true environment index $e$ is not provided in training, unless otherwise stated.

In this work, we assume an underlying structural causal model (SCM) governing the data generation process [35, 37]. An SCM considers a set of variables associated with vertices of a directed acyclic graph, where directed edges represent direct causation. Each variable is obtained as a result of an assignment of a deterministic function depending on the parental variables in the graph and an exogenous random variable. As an example, we present the following data generation process that is also considered in [3, 4]:

$$Y^e = g_v(\boldsymbol{X}_v^e, \epsilon_v), \;\; \boldsymbol{X}_v^e \perp \epsilon_v; \qquad \boldsymbol{X}^e = q(\boldsymbol{X}_v^e, \boldsymbol{X}_s^e), \tag{1}$$

where $g_v$ denote a non-degenerate deterministic function and $\epsilon_v$ is an independent noise variable. We observe data $\{\boldsymbol{X}^e, Y^e\}_{e=1}^E$ from multiple environments, each with probability $P(\boldsymbol{X}^e, Y^e)$. Then the marginal distribution of the mixed data can be represented as $P(\boldsymbol{X}, Y) = \sum_{e=1}^E \alpha_e P(\boldsymbol{X}^e, Y^e)$ for some $\alpha_e > 0$ and $\sum_{e=1}^E \alpha_e = 1$.

We consider the model as the composition of a feature extractor $\Phi$ and a classifier (or predictor) $f_\omega$ that is parameterized by $\omega$. To seek for out-of-distribution generalization ability, we hope that $\Phi$ merely encodes the information of invariant features. Let $\ell(\cdot, \cdot)$ denote a loss function such as cross-entropy loss and squared error. Our goal is to learn a robust and invariant model that minimizes the following objective over all the considered environments $e \in \mathcal{E}_{supp}$:

$$\sup_{e \in \mathcal{E}_{supp}} \mathbb{E}_{\boldsymbol{X}^e, Y^e} \left[ \ell \left( f_\omega(\Phi(\boldsymbol{X}^e)), Y^e \right) \right].$$

While there is no single, common definition of *invariant feature* in the literature, we treat the direct causal parents of $Y$ as invariant features, as in [36, 4]. This is because the conditional probability of the outcome $Y$ given its parental variables or direct causes remains unchanged with any intervention (not on $Y$). In contrast, if a non-parental variable is included in the conditioned variables, then the conditional probability could change after some intervention. As such, the underlying SCM governing data generation determines the desired invariance.

## 4 Impossibility Result

The first question we ask is whether it is possible to learn invariant models from the heterogeneous data of multiple environments with unknown environmental indexes. The following example immediately gives a negative answer.

**Example.** Consider that there are two environments $e \in \{1, 2\}$ with $\alpha_1 = \alpha_2 = 0.5$, and we observe binary-valued features $X_1, X_2$ and label $Y$. Suppose the joint distribution of mixed environments is given in Eq. (2). A learning algorithm that tries to learn invariant feature based on this dataset would result in a model that (deterministically) depends on either $X_1$, $X_2$, or both to predict $Y$. However, as discussed in last section, whether the learned features are invariant

$$
\begin{cases}
Y = 0, & w.p.\ 0.5, \\
Y = 1, & w.p.\ 0.5, \\
X_2 = X_1 = Y, & w.p.\ 0.6375, \\
X_2 \neq X_1 = Y, & w.p.\ 0.1125, \\
X_1 \neq X_2 = Y, & w.p.\ 0.2125, \\
X_2 = X_1 \neq Y, & w.p.\ 0.0375.
\end{cases}
\tag{2}
$$

is determined by the underlying data generation process. We now present two possible data generation processes that generate the same distribution yet have different invariant features:

- $X_1$ is the invariant feature while $X_2$ is spurious, where $p_s^{e=1} = 0.8$ and $p_s^{e=2} = 0.9$:

$$
X_1 \sim \text{Bernoulli}(0.5), \quad Y = \begin{cases} X_1, & w.p.\ 0.75, \\ 1 - X_1, & w.p.\ 0.25, \end{cases} \quad X_2^e = \begin{cases} Y, & w.p.\ p_s^e, \\ 1 - Y, & w.p.\ 1 - p_s^e. \end{cases}
\tag{3}
$$

- $X_2$ is the invariant feature while $X_1$ is spurious where $p_s^{e=1} = 0.8$ and $p_s^{e=2} = 0.7$:

$$
X_2 \sim \text{Bernoulli}(0.5), \quad Y = \begin{cases} X_2, & w.p.\ 0.85, \\ 1 - X_2, & w.p.\ 0.15, \end{cases} \quad X_1^e = \begin{cases} Y, & w.p.\ p_s^e, \\ 1 - Y, & w.p.\ 1 - p_s^e. \end{cases}
\tag{4}
$$

The joint distribution of the mixed environments for each data generation process is consistent with Eq. (2). Thus, from the joint distribution $P(X_1, X_2, Y)$, a learned model only depends on either $X_1$, $X_2$ or both, and fails to generalize for at least one of the two data generation processes. On the other hand, when the partition is given, one can verify that IRM would correctly identify $X_1$ (resp. $X_2$) as the invariant feature for the first (resp. second) scenario.

The above toy example is inspired by commonly-used classification tasks in the IRM literature, e.g., CMNIST [4] and CifarMnist [26]. For instance, in CMNIST, invariant feature $X_1$ denotes the semantic feature of the shape of hand-written digits '0' and '1', and spurious feature $X_2$ represents the color, which is either red or green. Label $Y \in \{0, 1\}$ corresponds to the digit shape and is also binary. Indeed, the construction procedure of CMNIST in [4] can be exactly described by the data generation process in Eq. (3). To further demonstrate the impossibility result, we next conduct an empirical validation on a new dataset in accordance to the data generation process in Eq. (4)

**Empirical Validation.** We construct a variant of CMNIST according to the second data generation process. In this new dataset, color is the invariant feature and digit shape is spurious. We use the same notations where $X_1$ stands for digit shape, $X_2$ is the color, and $Y$ denotes label. As described in Eq. (4), the label corresponds to the digit color and the digit shape is spuriously correlated with the label. We name this variant of CMNIST as MCOLOR (short for MNIST-COLOR). In the test domain, we set $p_s^{e=3} = 0.1$ to simulate the distributional shift, same as in CMNIST. We can then compare ERM, EIIL, IRM with environment partition, and LfF (learning from failures) [34] that tries to learn a robust model by boosting from wrongly specified samples of shallow neural networks. The empirical results are reported in Table 1. For ERM (oracle), we train the model only on the invariant feature, i.e., digit shape in CMNIST and color in MCOLOR.

We can see that the EIIL method performs poorly on the MCOLOR dataset. This is due to the inductive bias of EIIL and it indeed relies on the digit shape as the invariant feature. Since we have no prior knowledge of the data generation process, the true invariant feature can be the color, e.g., in the MCOLOR dataset. Similarly, ERM and LfF all rely on either color or shape as the invariant feature and would fail on at least one of CMNIST and MCOLOR. On the other hand, if environment partition is available, IRM can still learn the desired invariant feature.

**Table 1:** Experimental results on CMNIST and MCOLOR.

| Method | Env Partition | CMNIST | | MCOLOR | |
|---|---|---|---|---|---|
| | | Train Acc | Test Acc | Train Acc | Test Acc |
| ERM (oracle) | No | 75.2±0.2 | 72.1±0.1 | 85.0±0.0 | 85.0±0.0 |
| ERM | No | 86.4 ±0.0 | 14.5 ±0.1 | 86.3±0.1 | 80.1±0.6 |
| IRM | Yes | 71.4 ±0.3 | 66.4 ±0.3 | 84.9±0.1 | 84.7±0.3 |
| EIIL | No | 72.5 ±0.7 | 67.2 ±3.3 | 74.0±0.4 | 17.8±0.4 |
| LfF | No | 76.7 ±0.3 | 21.2 ±0.4 | 76.6±0.0 | 74.2±0.0 |

Moreover, such examples are not rare. For any data distribution $P(\boldsymbol{X}, Y)$ generated by some data generation process like (1) (which can be replaced by other forms of SCM), we can find a different data generation process that induces the same distribution yet has different invariant/spurious features. Similarly, it is impossible to identify the spurious features only based on $P(\boldsymbol{X}, Y)$. This result is formally summarized in Theorem 1.

**Theorem 1.** *Let $\boldsymbol{X}_v$ and $\boldsymbol{X}_s$ be respectively the invariant and spurious features, with label $Y$. For the joint distribution $P(\boldsymbol{X}, Y)$ consisting of data from multiple environments with $P(\boldsymbol{X}_s, Y) \neq 0$, we can find $\boldsymbol{X}'_v$ and $\boldsymbol{X}'_s$ as invariant and spurious features so that:*

- $\boldsymbol{X}'_v \neq \boldsymbol{X}_v$ *and* $\boldsymbol{X}'_v \cap \boldsymbol{X}_s \neq \varnothing$*;*
- $(\boldsymbol{X}'_v, \boldsymbol{X}'_s)$ *together with some noise variables generate the same distribution* $P(\boldsymbol{X}, Y)$*.*

A proof is provided in Appendix B.1. Under this circumstance, we have to introduce additional assumptions/conditions or certain "inductive bias" to identify the invariant features [45, 37, 20, 32, 50]. The latter may come implicitly from the proposed algorithm, model, and/or the data. We believe that this is the reason that makes existing methods achieve improved empirical performance in their considered scenarios. For example, in [29, 30], the discrepancy of spurious features among clusters are expected to be larger than that of causal features, and in [10, 34, 42, 28], the ERM model of the first stage should heavily or fully rely on the spurious feature. However, these inductive biases may not be always guaranteed. Thus, as suggested by Theorem 1, the role of inductive biases shall be discussed more explicitly when learning invariant models without environment partition.

Alternatively, in this work, we consider that there exists additionally observed variable $\boldsymbol{Z}$ with the data, which has been often considered in the literature [19, 20, 22, 53, 51]. The variable $\boldsymbol{Z}$ could be, for example, the time index in a time series, some kind of class labels, concurrently observed variables, or human annotations on potential unmeasured variable [46]. It is worth noting that Xie et al. [53] also take advantage of auxiliary information to help improve OOD performance in a semi-supervised manner. However, they consider a different setting with access to unlabeled (out-of-domain) test data together with additional unlabeled training data. In the next section, we show how to utilize $\boldsymbol{Z}$ to learn invariant models from the heterogeneous dataset.

## 5   ZIN: Learning Invariance with Additional Auxiliary Information

We consider that there exists additional auxiliary information $\boldsymbol{Z} \in \mathbb{R}^{d_z}$ in company with the data $(\boldsymbol{X}, Y)$. In this section, we propose ZIN, *auxiliary information $\boldsymbol{Z}$ for environmental INference*, for invariant learning from the heterogeneous dataset $\mathcal{D}$ without environment partition. We will derive conditions for invariance identification and show these conditions are both sufficient and necessary in a fairly general setting.

### 5.1   Method

We aim to learn a function $\rho(\cdot) : \mathbb{R}^{d_z} \to \mathbb{R}^K$ that softly assigns a sample to $K$ environments. Here $K$ is a pre-specified number (a hyper-parameter) and its choice will be empirically investigated in our experiments. Let $\rho^{(k)}(\cdot)$ denote the $k$-th entry of $\rho(\cdot)$, with $\rho(\boldsymbol{Z}) \in [0,1]^K$ and $\sum_k \rho^{(k)}(\boldsymbol{Z}) = 1$. Denote the ERM loss as $\mathcal{R}(\omega, \Phi) = \frac{1}{n} \sum_{i=1}^n \ell\left(f_\omega(\Phi(\boldsymbol{x}_i)), y_i\right)$ and the loss in the $k$-th inferred environment as $\mathcal{R}_{\rho^{(k)}}(\omega, \Phi) = \frac{1}{n} \sum_{i=1}^n \rho^{(k)}(\boldsymbol{z}_i)\ell(f_\omega(\Phi(\boldsymbol{x}_i)), y_i)$.

Recall that IRM [4] learns an invariant representation $\Phi$, upon which there is a classifier $f_\omega$ that is simultaneously optimal in all environments. Suppose that the environments have been given according to a fixed $\rho(\cdot)$. To measure the optimality of $f_\omega$ in the $k$-th environment, we can fit an environment-dependent classifier $f_{\omega_k}$ on the data from that environment. If $f_{\omega_k}$ achieves a smaller loss, then we know that $f_\omega$ is not optimal in this environment. We can further train a set of environment-dependent classifiers $\{f_{\omega_k}\}_{k=1}^K$, one for each environment, to measure whether $f_\omega$ is simultaneously optimal in all environments. Thus, when $\rho(\cdot)$ is provided, our formulation to learn invariance is as follows:

$$\min_{\omega,\Phi} \max_{\{\omega_k\}} \mathcal{L}(\Phi, \omega, \omega_1, \cdots, \omega_K, \rho) := \mathcal{R}(\omega, \Phi) + \lambda \underbrace{\sum_{k=1}^K \left[ \mathcal{R}_{\rho^{(k)}}(\omega, \Phi) - \mathcal{R}_{\rho^{(k)}}(\omega_k, \Phi) \right]}_{\text{invariance penalty}}. \quad (5)$$

If $\Phi$ extracts spurious features that are unstable in the inferred environments, $\mathcal{R}_{\rho^{(k)}}(\omega, \Phi)$ will be larger than $\mathcal{R}_{\rho^{(k)}}(\omega_k, \Phi)$, resulting in a non-zero invariance penalty.

Next, we consider how to learn the partition function $\rho(\cdot)$. A *good* partition function should generate environments among which the spurious features exhibit instability, so that there is a large penalty if $\Phi$ extracts spurious features. Thus, we seek for an environment partition that maximizes the invariance penalty. The overall framework is provided below:

$$\min_{\omega,\Phi} \max_{\rho,\{\omega_1,\cdots,\omega_K\}} \mathcal{L}(\Phi, \omega, \omega_1, \cdots, \omega_K, \rho). \quad (6)$$

The idea of our method can be summarized as inferring an environment partition and then learning invariant models based on the inferred environments. Creager et al. [10] has used a similar intuition and adopted a pre-trained biased model to estimate the environments. However, their two-stage method cannot be jointly optimized, and the environment partition relies on the given model and lacks a theoretical guarantee. In contrast, we will first present sufficient conditions on $\boldsymbol{Z}$ so that the proposed framework can provably identify the invariant features. We further show that these conditions are also necessary, and that violation of these conditions will lead to failure of invariance identification. Assigning independent weights to each data sample, which is adopted in EIIL [10], is unfortunately an instance of such violations and may fail to identify invariant features.

### 5.2 Sufficient Conditions for Identifiability

In this section, we try to understand ZIN from a theoretical perspective. We start with a simple yet general setting: $\boldsymbol{X} = [\boldsymbol{X}_v; \boldsymbol{X}_s]$ (i.e., no scramble on the observation), $\Phi \in \{0,1\}^d$ is an element-wise feature selection mask, and $f_\omega$ is a general non-linear function. We also focus on classification tasks with $\ell(\cdot, \cdot)$ being the cross-entropy loss, as there is an interesting connection to (conditional) Shannon entropy. Extensions to other loss functions and linear feature transformations are left to Appendix A.1 and A.2, respectively.

In this setting, our goal is equivalent to learning the optimal feature mask that merely selects invariant features, i.e., $\Phi_v = [\mathbf{1}^{d_v}; \mathbf{0}^{d_s}]$. For a given feature mask $\Phi$, the objective function is equal to $\hat{\mathcal{L}}(\Phi) = \min_\omega \max_{\rho,\{\omega_k\}} \mathcal{L}(\Phi, \omega, \omega_1, \cdots, \omega_K, \rho)$. Then ZIN can correctly identify the invariant features, or solution to Problem 6 is equivalent to $\Phi_v$, if and only if $\hat{\mathcal{L}}(\Phi_v) < \hat{\mathcal{L}}(\Phi)$ for all $\Phi \neq \Phi_v$. This observation will be used to establish our main theoretic result.

With a little abuse of notation, we use $H(Y|\boldsymbol{X}')$ to denote expected loss of an optimal classifier over some $\boldsymbol{X}'$ and $Y$, and similarly $H(Y|\rho(\boldsymbol{Z}), \boldsymbol{X}')$ to denote the minimum risk $\sum_{k=1}^K \mathcal{R}_{\rho^{(k)}}(\omega_k, \Phi)$ for a given $\rho(\boldsymbol{Z})$. With cross-entropy loss and when $\rho(z_i)$ gives exactly one environment, i.e., $\rho(\boldsymbol{Z})$ is one-hot, it can be verified that the optimal loss $H(\cdot|\cdot)$ coincides with conditional entropy. In the following we first state the assumptions for our identifiabitiliy result.

**Assumption 1.** *For a given feature mask $\Phi$ and any constant $\epsilon > 0$, there exists $f \in \mathcal{F}$ such that $\mathbb{E}[\ell(f(\Phi(\boldsymbol{X})), Y)] \leq H(Y|\Phi(\boldsymbol{X})) + \epsilon$.*

**Assumption 2.** *If a feature violates the invariance constraint, adding another feature would not make the penalty vanish, i.e., there exists a constant $\delta > 0$ so that for spurious feature $\boldsymbol{X}_1 \subset \boldsymbol{X}_s$ and any feature $\boldsymbol{X}_2 \subset \boldsymbol{X}$, $H(Y|\boldsymbol{X}_1, \boldsymbol{X}_2) - H(Y|\rho(\boldsymbol{Z}), \boldsymbol{X}_1, \boldsymbol{X}_2) \geq \delta(H(Y|\boldsymbol{X}_1) - H(Y|\rho(\boldsymbol{Z}), \boldsymbol{X}_1))$.*

**Assumption 3.** *For any distinct features $\boldsymbol{X}_1$, $\boldsymbol{X}_2$, $H(Y|\boldsymbol{X}_1, \boldsymbol{X}_2) \leq H(Y|\boldsymbol{X}_1) - \gamma$ with fixed $\gamma > 0$.*

Assumption 1 is a common assumption that requires the function space $\mathcal{F}$ be rich enough such that, given $\Phi$, there exists $f \in \mathcal{F}$ that can fit $P(Y|\Phi(\boldsymbol{X}))$ well. Assumption 2 aims to ensure a sufficient positive penalty if a spurious feature is included (see Appendix A.3 for further discussion). Assumption 3 indicates that any feature contains some useful information w.r.t. $Y$, which cannot be explained by other features. Otherwise, we can simply remove such a feature (e.g., by variable selection methods [49]), as it does not affect prediction. We next present our sufficient conditions for ZIN to identify invariant features.

**Condition 1** (Invariance Preserving Condition). *Given invariant feature $\boldsymbol{X}_v$ and any function $\rho(\cdot)$, it holds that $H(Y|\boldsymbol{X}_v, \rho(\boldsymbol{Z})) = H(Y|\boldsymbol{X}_v)$.*

**Condition 2** (Non-invariance Distinguishing Condition). *For any feature $\boldsymbol{X}_s^k \in \boldsymbol{X}_s$, there exists a function $\rho(\cdot)$ and a constant $C > 0$ such that $H(Y|\boldsymbol{X}_s^k) - H(Y|\boldsymbol{X}_s^k, \rho(\boldsymbol{Z})) \geq C$.*

We remark that Condition 1 can be met if $H(Y|\boldsymbol{X}_v, \boldsymbol{Z}) = H(Y|\boldsymbol{X}_v)$ (a proof is provided in Appendix B.6). Condition 1 requires that invariant features should remain invariant w.r.t any environment partition induced by $\rho(\boldsymbol{Z})$. Otherwise, if there exists a partition where an invariant feature becomes non-invariant, then this feature would induce a positive penalty. Condition 2 implies that for each spurious feature, there exists at least one partition so that this feature is non-invariant in the split environments. If a spurious feature does not incur any invariance penalty in all possible environment partitions, we can never distinguish it from true invariant features. With these conditions, our main result follows and a proof is given in Appendix B.2.

**Theorem 2** (Identifiability of Invariant Features). *With Assumptions 1-3 and Conditions 1-2, if $\epsilon < \frac{C\gamma\delta}{4\gamma + 2C\delta H(Y)}$ and $\lambda \in [\frac{H(Y)+1/2\delta C}{\delta C - 4\epsilon} - \frac{1}{2}, \frac{\gamma}{4\epsilon} - \frac{1}{2}]$, then we have $\hat{\mathcal{L}}(\Phi_v) < \hat{\mathcal{L}}(\Phi)$ for all $\Phi \neq \Phi_v$, where $H(Y)$ denotes the entropy of $Y$. Thus, the solution to Problem 6 identifies invariant features.*

### 5.3 Necessary Conditions for Identifiability

Conditions 1 and 2 may appear rather strong. In this section, we prove that they are also necessary conditions for ZIN to identify invariant features. Specifically, Proposition 1 shows that if Condition 1 is violated, then some invariant features will be excluded in the solution of Problem 6; and in Proposition 2, violation of Condition 2 renders some spurious features included in the solution.

**Proposition 1.** *With Assumptions 1-3, if Condition 1 is violated, i.e., there exists $\rho(\cdot)$ so that $H(y|\boldsymbol{X}_v) - H(y|\boldsymbol{X}_v, \rho(\boldsymbol{Z})) \geq C' > 0$, then there exists a mask $\Phi' \neq \Phi_v$ with $\hat{\mathcal{L}}(\Phi_v) > \hat{\mathcal{L}}(\Phi')$.*

A proof is provided in Appendix B.3, which is similar to the first step of the proof of Theorem 2. A question is how Condition 1 could be violated. On the other hand, while we introduce $\boldsymbol{Z}$ as additional variables, it is also interesting to know whether we can obtain a valid partition based on only the training data $(\boldsymbol{X}, Y)$. Below we provide examples regarding these questions, with proofs given in Appendix B.4. As $h(\text{Index}(\boldsymbol{X}, Y))$ can be treated as learning independent weights for each sample based on $\{(\boldsymbol{x}_i, y_i)\}_i$, this case includes EIIL [10] as an instance.

**Corollary 1.** *Condition 1 is violated for the following cases: there exists a function $\rho(\cdot)$ and an injective function $h(\cdot)$ so that (a) $\rho(\boldsymbol{Z}) = h(Y)$, (b) $\rho(\boldsymbol{Z}) = h(\boldsymbol{X}, Y)$, or (c) $\rho(\boldsymbol{Z}) = h(\text{Index}(\boldsymbol{X}, Y))$.*

We now discuss the necessity of Condition 2, with an additional assumption.

**Assumption 4.** *If two features are invariant w.r.t. an environment partition, then the concatenated features are also invariant. That is, for $\boldsymbol{X}_1, \boldsymbol{X}_2 \subset \boldsymbol{X}$ and $\rho(\cdot)$, if $H(Y|\boldsymbol{X}_1) - H(Y|\rho(\boldsymbol{Z}), \boldsymbol{X}_1) = 0$ and $H(Y|\boldsymbol{X}_2) - H(Y|\rho(\boldsymbol{Z}), \boldsymbol{X}_2) = 0$, we have $H(Y|\boldsymbol{X}_1, \boldsymbol{X}_2) - H(Y|\rho(\boldsymbol{Z}), \boldsymbol{X}_1, \boldsymbol{X}_2) = 0$.*

**Proposition 2.** *With Assumptions 1, 3 and 4, if Condition 2 is violated, i.e., there exists a spurious feature $\boldsymbol{X}_s^k \in \boldsymbol{X}_s$ such that $H(Y|\boldsymbol{X}_s^k) - H(Y|\boldsymbol{X}_s^k, \rho(\boldsymbol{Z})) = 0$ for any $\rho(\cdot)$, then there exists a feature mask $\Phi' \neq \Phi_v$ with $\hat{\mathcal{L}}(\Phi_v) > \hat{\mathcal{L}}(\Phi')$.*

Since there exists a spurious feature $\boldsymbol{X}_s^k$ that is "invariant" in all possible environment partition, adding this feature to $\Phi_v$ does not induce any invariance penalty but increases the prediction power.

Thus, such a feature mask can achieve a smaller loss than $\hat{\mathcal{L}}(\Phi_v)$. For completeness, we provide a proof in Appendix B.5. Proposition 2 again indicates in theory that $\mathbf{Z}$ should be sufficiently diverse and informative so that each spurious feature can be recognized.

## 5.4 Choice of the Auxiliary Information

Conditions 1 and 2 present theoretical requirements of $\mathbf{Z}$ to successfully learn invariant features. We now discuss how to find such auxiliary information given certain prior knowledge.

Take Fig. 1 for example. When we collect a data point (e.g., taking a photo), there often exists some meta information (e.g., time slot, coordinate, and temperature). Suppose that the image is generated according to the causal graph on the right panel of Fig. 1. In this case, the invariant feature $\mathbf{X}_v$ consists of $X_1$ and $X_2$, and the meta information can be used as $\mathbf{Z}$ which is colored in green in the causal graph. This is because $H(Y|X_1, X_2) = H(Y|X_1, X_2, \mathbf{Z})$ or equivalently $Y \perp \mathbf{Z} \mid X_1, X_2$, so that Condi-

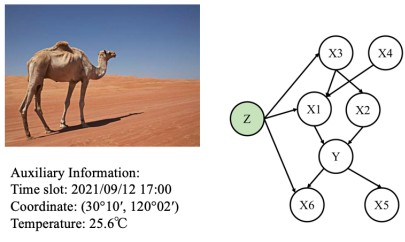

Auxiliary Information:
Time slot: 2021/09/12 17:00
Coordinate: (30°10′, 120°02′)
Temperature: 25.6℃
...

**Figure 1:** An example of $Z$ satisfying Condition 1.

tion 1 holds. This example shows some evidence of choosing the auxiliary information: $\mathbf{Z}$ and $Y$ should be d-separated by $\mathbf{X}_v$. There are other examples satisfying Condition 1, which are given Fig. 5 in Appendix E. We also illustrate some failure choices in Fig. 6 in Appendix E. Specifically, if $\mathbf{Z}$ is a child or a direct cause of $Y$, then it contains additional information of $Y$ conditional on the invariant feature, violating Condition 1. This kind of information cannot be used in our framework.

To summarize, with some prior knowledge on the path between $\mathbf{Z}$ and $Y$, we suggest to choose as much auxiliary information as possible under the following condition: *A feasible choice of $\mathbf{Z}$ should satisfy Condition 1 and should be correlated with certain variables in the causal graph of the SCM. The path between $\mathbf{Z}$ and $Y$ should be d-separated by $\mathbf{X}_v$ (e.g., in Figs. 1 and 5) or there is no path between $\mathbf{Z}$ and $Y$. Notably, $\mathbf{Z}$ cannot be the parent or child of $Y$ as shown in Fig. 6.*

# 6 Experiments

This section empirically verifies our theoretic analysis and the effectiveness of ZIN on both synthetic and real world datasets. We compare ZIN with several existing methods: ERM, IRM, [4], group DRO [41], EIIL [10], HRM [29], and LfF [34]. We provide the ground-truth partition to IRM and group DRO. Notice that LfF only works for classification tasks.

We implement both $\rho$ and $f$ in ZIN as a two-layer MLP with 32 hidden units. We adopt the first order approximation of Eq. (5), as described in Appendix C. The number of inferred environments $K$ is set to be 2 as default. We implement $\Phi$ as a two-layer MLP for the synthetic dataset and house price prediction, ResNet-18 [16] for CelebA, and 1D CNN for Landcover. More details are provided in Appendix C.

## 6.1 Synthetic Dataset

**Temporal Heterogeneity**. We consider temporal heterogeneity with distributional shift w.r.t. time. Let $t \in [0, 1]$ be time index and $X_v(t) \in \mathbb{R}$ the invariant feature. The data generation process is

$$X_v(t) \sim \begin{cases} \mathcal{N}(1, \sigma^2), & w.p.\ 0.5, \\ \mathcal{N}(-1, \sigma^2), & w.p.\ 0.5, \end{cases} \quad Y(t) \sim \begin{cases} \text{sign}(X_v(t)), & w.p.\ p_v, \\ -\text{sign}(X_v(t)), & w.p.\ p_v', \end{cases} \quad X_s(t) \sim \begin{cases} \mathcal{N}(Y(t), \sigma^2), & w.p.\ p_s(t), \\ \mathcal{N}(-Y(t), \sigma^2), & w.p.\ p_s'(t), \end{cases}$$

where $p_v$ is a constant w.r.t. $t$, indicating a stable correlation between $Y(t)$ and $X_v(t)$, $p_v' = 1 - p_v$, and $p_s'(t) = 1 - p_s(t)$. Notice here $p_s(t)$ varies with time $t$. A similar setting with two-dimensional spatial variable is considered in Appendix D.1. Our goal is to learn a model that purely relies on $X_v$. We simulate two heterogeneous environments along time, namely, $\{[0, 0.5], [0.5, 1]\}$, and $p_s(t)$ will be set differently. We use tuple of $p_s(t)$ in the two environments to denote a simulated case. For example, $(0.999, 0.7)$ stands for $p_s(t) = 0.999, t \in [0, 0.5)$ and $p_s(t) = 0.7, t \in [0.5, 1]$. We

**Table 2:** Test Mean and Worst accuracy (%) on four temporal heterogeneity synthetic datasets.

| Env Partition | $p_s(t)$ | 0.999, 0.7 | | | | 0.999, 0.8 | | | | 0.999, 0.9 | | | |
|---|---|---|---|---|---|---|---|---|---|---|---|---|---|
| | $p_v$ | 0.9 | | 0.8 | | 0.9 | | 0.8 | | 0.9 | | 0.8 | |
| | Test Acc | Mean | Worst | Mean | Worst | Mean | Worst | Mean | Worst | Mean | Worst | Mean | Worst |
| No | ERM | 75.37 | 57.31 | 59.65 | 25.81 | 68.72 | 41.97 | 55.90 | 15.07 | 60.61 | 23.39 | 52.85 | 7.57 |
| | EIIL | 38.41 | 16.80 | 64.89 | 49.15 | 50.77 | 46.67 | 68.36 | 56.35 | 61.99 | 53.81 | 70.10 | 59.36 |
| | HRM | 50.00 | 49.99 | 49.98 | 49.93 | 50.00 | 49.98 | 50.01 | 49.99 | 50.00 | 49.98 | 49.99 | 49.97 |
| | ZIN | **87.50** | **85.36** | **77.85** | **75.39** | **86.35** | **82.91** | **76.79** | **72.77** | **83.71** | **75.89** | **73.55** | **64.69** |
| Yes | IRM | 87.57 | 85.47 | 77.99 | 75.65 | 86.57 | 83.25 | 77.00 | 73.39 | 83.99 | 76.48 | 73.84 | 65.33 |

evaluate the performance on four distinct test environments with $p_s \in \{0.999, 0.8, 0.2, 0.1\}$ and $p_v$ being constant. We use $t$ as the auxiliary information. More details are provided in Appendix C

**Results**. Table 2 reports the test accuracy. In all simulated settings, the worst accuracy of ERM is much lower than the mean accuracy, indicating that ERM tends to rely on spurious feature $X_s$. EIIL can improve the worst accuracy in some cases, e.g., when $p_s(t) = (0.999, 0.8)$ and $p_s(t) = (0.999, 0.9)$. However, its performance is even worse than ERM for some other settings. This may be attributed to the first stage of EIIL, where the trained biased model is not guaranteed and may learn both spurious and invariant features. The proposed method ZIN improves the worst test accuracy significantly. For instance, when $p_s(t) = (0.999, 0.7)$ and $p_v = 0.9$, ZIN outperforms ERM and EIIL by over 28% and 65%, respectively. Moreover, ZIN is very close to IRM that knows the ground-truth environment partition, showing that ZIN can infer the environments effectively. Finally, it seems that HRM does not learn a useful model for this classification task.

**Ablation Study**. In Appendix D.2, we conduct ablation study to verify our theoretical results in Section 5 by choosing different auxiliary variables. We also empirically study the choice of hyper-parameter $K$. A notable observation is that $K$ has a relatively small impact, as shown in Fig. 3.

## 6.2 Real World Datasets

**House Price Prediction**. This experiment considers a real world regression dataset of house sales prices from Kaggle (https://www.kaggle.com/c/house-prices-advanced-regression-techniques). The target variable is house price and each sample contains 17 predictive variables like the built year of the house, number of bedrooms, etc. The dataset is split according to the built year, with training dataset in period $[1900, 1950]$ and test dataset in period $(1950, 2000]$. We normalize the prices of the houses with the same built year, and our target is to predict the normalized price. We choose the built year as auxiliary information for ZIN. As there is no well-defined ground-truth partition for IRM, we manually split the training dataset equally into 5 segments with 10-year range in each segment.

Table 3 reports the mean squared error (MSE), and ZIN again outperforms other methods on test dataset. We also investigate whether this type of additional information $Z$, together with $X$, helps EIIL and HRM. The corresponding results are marked with $(+Z)$ in Table 3. We observe that the results of EIIL and HRM with additional information $Z$ are worse than those without using $Z$. This implies that such information may not be directly utilized by or even can harm the two methods.

**CelebA**. This task is to predict *Smiling* based on the image from CelebA [31], and by construction the target is spuriously correlated with *Gender*. We have access to the meta annotations of CeleA images and they serve as $Z$ in our framework. In particular, we pick seven features as potential auxiliary variables: {*Young*, *Blond Hair*, *Eyeglasses*, *High Cheekbones*, *Big Nose*, *Bags Under Eyes*, *Chubby*}. Notice that the annotation of *Gender* is not provided to ZIN, EIIL, HRM and ERM, but is made available to IRM and group DRO to create an oracle environment partition.

Empirical results are given in Table 4, where ZIN(#) represents ZIN using # of the seven features as $Z$. ERM achieves the highest training accuracy, while only has 47.58% worst test accuracy. The worst test performances of ERM, EIIL and LfF indicate that they may not learn the causal features. Finally, we observe that ZIN performs better when more auxiliary information is provided; specifically, ZIN(7) achieves the best test performance based on the mixed dataset. This observation validates our discussion on choosing the auxiliary information in Section 5.4.

**Table 3:** House price prediction (MSE).

| Method | Env Index | Train | Test Mean | Test Worst |
|--------|-----------|-------|-----------|------------|
| IRM | Yes | 0.1327 | 0.4456 | 0.6821 |
| group DRO | Yes | 0.1213 | 0.6887 | 1.0050 |
| ERM | No | **0.1141** | 0.4764 | 0.6703 |
| EIIL | No | 0.6841 | 0.9625 | 1.3909 |
| EIIL(+$Z$) | No | 0.6912 | 0.9701 | 1.4201 |
| HRM | No | 0.3466 | 0.4621 | 0.5721 |
| HRM(+$Z$) | No | 0.3190 | 0.4221 | 0.5873 |
| ZIN | No | 0.2275 | **0.3339** | **0.4815** |

**Table 4:** Accuracy (%) on CelebA task.

| Method | Env Index | Train | Test Mean | Test Worst |
|--------|-----------|-------|-----------|------------|
| IRM | Yes | 81.30±1.53 | 78.44±0.48 | 75.03±1.29 |
| group DRO | Yes | 89.32±0.67 | 74.28±0.22 | 58.11±0.61 |
| ERM | No | **90.97**±0.66 | 70.76±0.26 | 47.58±0.46 |
| LfF | No | 59.89±0.72 | 52.97±0.56 | 44.38±2.01 |
| EIIL | No | 90.01±0.73 | 71.45±0.29 | 50.48±1.98 |
| ZIN(1) | No | 90.62±0.78 | 70.79±0.61 | 47.62±0.98 |
| ZIN(4) | No | 83.57±1.40 | 75.20±0.71 | 63.47±1.41 |
| ZIN(7) | No | 83.06±1.28 | **76.29**±0.60 | **67.27**±1.15 |

In Appendix D, Fig. 3 reports the results of different choices of hyper-parameter $K$ and Fig. 4 visualizes the inferred environments in this experiment. Again, we find that $K$ has a relatively small impact on the proposed algorithm.

**Landcover**. Our final task is land cover prediction that classifies the land cover type (e.g., grasslands) from satellite data [14, 40]. We take the same setup from [53]: the time series data input dimension is $46 \times 8$; the target $Y$ is one of six land cover classes; six climate related variables are the auxiliary variables. We also consider non-African locations as training data and Africa as test data. In addition, we take location (latitude and longitude) as possible ad-

**Table 5:** Test Accuracy (%) on Landcover task.

| Method | IID Test | OOD Test |
|--------|----------|----------|
| ERM [52] | 75.92 | 58.31 |
| ERM (+climate) [52] | 76.58 | 54.78 |
| LfF | 66.24 | 61.69 |
| EIIL | 72.61 | 64.79 |
| ZIN (climate) | 72.56 | 62.50 |
| ZIN (location) | 72.18 | **66.06** |

ditional information. For ZIN, we average climate variables over the time dimension and use those means to predict environments. In Table 5, we see that ZIN with location achieves the best OOD performance, while ZIN with climate variables is slightly worse. This is because climate variables indeed contain some information for predicting the target, violating our conditions on $Z$. EIIL also has a good performance in this task, and we conjecture that its first stage has learned the spurious features as desired. However, together with previous empirical results, it can be risky as the first stage may not be guaranteed.

## 6.3 Discussion on Auxiliary Information for Related Methods

The proposed framework ZIN relies on additional auxiliary variables. It is interesting to ask whether this type of information is also useful to related methods such as EIIL and HRM. In many scenarios, the additional variable $Z$ may have no or little information about the target, e.g., 1) in time series tasks, the time index is rarely used to predict the label; and 2) in the CelebA classification task, features like *Young* and *Eyeglasses* are likely to provide little information about predicting *Smiling*. Then $Z$ may not be useful to ERM and EIIL. In Section 6.2, we also provide $Z$ to EIIL and HRM, and the experimental result shows that it does not help or even harms the test performance.

## 7 Concluding Remarks

This paper investigates when and how to learn invariance from heterogeneous data without explicit environment indexes. We first show that learning invariant models in this case is generally impossible. Then we propose ZIN to jointly learn environment partition and invariant representation using some additional auxiliary variables. We provide theoretic guarantees in both feature selection and linear feature learning scenarios. Experimental results verify our analysis and demonstrate an improved performance of ZIN over existing methods. Our results also raise the need of future works to make the role of inductive biases more explicit, when learning invariance from heterogeneous data without environment indexes. A limitation of the present work is the lack of theoretic guarantee with general nonlinear feature extractors, which is also an open problem for IRM even with environment partition.

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
