# A More Theoretical Results

## A.1 Extension to Other Loss Functions

Sections 5.2 and 5.3 focus on classification task with cross-entropy loss, to establish sufficient and necessary conditions for invariance identification. Similar results can be shown for other loss functions or tasks in a straightforward manner.

To see this, notice that $H(\cdot|\cdot)$ in Assumptions 1-3 and Conditions 1-2 is used to represent the optimal expected risk that coincides with the conditional entropy, with cross-entropy loss and when $\rho(z_i)$ gives exactly one environment. For other loss functions $l(\cdot, \cdot)$ like squared error, we can use $L(\cdot|\cdot)$ to represent the optimal expected risks, and $L(\cdot)$ denotes the optimal expected risk with no predictor variables, e.g., variance for the squared error loss. Replacing $H(\cdot|\cdot)$ with $L(\cdot|\cdot)$, we can follow the same proof procedure to obtain similar sufficient and necessary conditions. This is summarized in Theorem 3 for completeness.

**Theorem 3.** *Consider Assumptions 1-3 and Conditions 1-2 where $H(\cdot|\cdot)$ is replaced with $L(\cdot|\cdot)$ accordingly. If $\epsilon < \frac{C\gamma\delta}{4\gamma + 2C\delta L(Y)}$ and $\lambda \in [\frac{L(Y)+1/2\delta C}{\delta C - 4\epsilon} - \frac{1}{2}, \frac{\gamma}{4\epsilon} - \frac{1}{2}]$, we have $\hat{\mathcal{L}}(\Phi_v) < \hat{\mathcal{L}}(\Phi)$ for all $\Phi \neq \Phi_v$, where we assume $L(Y) < \infty$. If Condition 1 or Condition 2 is violated, then there exists a feature mask $\Phi' \neq \Phi_v$ so that $\hat{\mathcal{L}}(\Phi_v) > \hat{\mathcal{L}}(\Phi')$.*

## A.2 Beyond Feature Selection: Linear Feature Transformations

In Section 5, both the invariant and spurious features can be directly observed and we focus on the ability of ZIN to identify the invariant features. In this section, we extend feature selection to feature learning, and show that ZIN is able to learn the invariant features given a scrambled observation in a linear form following [4, 39]. Specifically, we consider the same data generation process as [4]:

$$Y^e = \boldsymbol{X}_v^e \cdot \beta + \epsilon_v, \quad \boldsymbol{X}_v^e \perp \epsilon_v, \quad \mathbb{E}[\epsilon_v] = 0; \qquad \boldsymbol{X}^e = \boldsymbol{W} \cdot [\boldsymbol{X}_v^e; \boldsymbol{X}_s^e], \tag{7}$$

where $\beta \in \mathbb{R}^{d_v}$ and $\boldsymbol{W} \in \mathbb{R}^{d \times (d_v + d_s)}$. We assume that there exists $\tilde{\boldsymbol{W}} \in \mathbb{R}^{d_v \times d}$ so that $\tilde{\boldsymbol{W}}(\boldsymbol{W}[\boldsymbol{x}_v; \boldsymbol{x}_s]) = \boldsymbol{x}_v$ for all $\boldsymbol{x}_v$ and $\boldsymbol{x}_s$. Both the feature extractor and predictor take a linear form, i.e., $\Phi$ takes values in $\mathbb{R}^{d \times d}$ and $\omega \in \mathbb{R}^d$. The prediction for $\boldsymbol{X}$ is $\omega \circ \Phi(\boldsymbol{X}) = (\Phi \boldsymbol{X})^T \omega$.

A major difficulty in this setting lies in how to characterize the effect of an invariant/spurious feature in a *quantitive* way: the feature extractor $\Phi$ may extract an arbitrarily small portion of spurious information. Following [4, 39], we consider a constrained form of Problem 6 for theoretic analysis:

$$\min_{\omega, \Phi} \mathcal{R}(\omega, \Phi), \quad \text{subject to} \quad \max_{\rho, \{\omega_k\}} \sum_{k=1}^{K} \left[ \mathcal{R}_{\rho^{(k)}}(\omega, \Phi) - \mathcal{R}_{\rho^{(k)}}(\omega_k, \Phi) \right] = 0. \tag{8}$$

As in Conditions 1 and 2, the auxiliary information should also be sufficiently informative so that the inferred environments can be diverse enough but also maintain the underlying invariance. This is in accordance to existing conditions for identifiability in the linear case [4, 39]. In this paper, we utilize such a condition, *linear general position* condition, from [4]. For our analysis, we take squared error as our loss function and consider that $\rho(\cdot)$ partitions the environments in a hard manner, i.e., each data sample would be assigned to exactly one environment. As in Appendix A.1, we use $L(\cdot|\cdot)$ to represent the optimal expected risks. We also assume that the environments are non-degenerate, i.e., each inferred environment contains some data samples; otherwise, we can simply remove such an environment. Our identifiability result for the linear case then follows.

**Proposition 3.** *Assume Condition 1 where $H(\cdot|\cdot)$ is replaced with $L(\cdot|\cdot)$ accordingly. Suppose that there exists $\rho(\cdot)$ such that the generated environments, denoted as $\{\boldsymbol{X}^k\}_{k=1}^{K}$, lie in linear general position of degree $r$, i.e., $K > d - r + d/r$ for some $r \in \mathbb{N}$ and for all non-zero $\boldsymbol{x} \in \mathbb{R}^d$: $\dim \left( \text{span} \left( \{\mathbb{E}_{\boldsymbol{X}^k} \left[ \boldsymbol{X}^k \boldsymbol{X}^{k^T} \right] \boldsymbol{x} - \mathbb{E}_{\boldsymbol{X}^k, \epsilon_v} \left[ \boldsymbol{X}^k \epsilon_v \right] \}_k \right) \right) > d - r$. If $\Phi \in \mathbb{R}^{d \times d}$ has rank $r > 0$, then Problem 8 results in the desired invariant predictor.*

*Proof.* **Step 1:** No spurious feature will be learned. Given a partition $\{\boldsymbol{X}^k\}_{k=1}^{K}$ that lies in linear general position of degree $r$, Arjovsky et al. [4, Theorem 9] shows that $\Phi$ and $\omega$ satisfies the normal equations $\Phi \mathbb{E}_{\boldsymbol{X}^k}[\boldsymbol{X}^k \boldsymbol{X}^{k^T}] \Phi^T \omega = \Phi \mathbb{E}_{\boldsymbol{X}^k, Y^k}[\boldsymbol{X}^k Y^k]$ for all $k$ if and only if $\Phi$ elicits the desired

invariant predictor $\Phi^T \omega = \tilde{W}^T \beta$. Thus, we only need to show our solution meets the same normal equations. Let $\Phi'$ and $\omega'$ denote a solution to Problem 8. According to the constraint and the general linear position condition, we know there exists a partition $\{X_k\}_{k=1}^K$ lies in general linear position of degree $r$, and we have $\mathcal{R}^k(\omega', \Phi') = \mathcal{R}^k(\omega_k, \Phi')$. Notice that $\omega_k$ minimizes the mean squared error on only the $k$-th environment, hence it must satisfy the normal equation $\Phi' \mathbb{E}_{X^k}[X^k X^{k^T}] \Phi'^T \omega_k = \Phi' \mathbb{E}_{X^k, Y^k}[X^k Y^k]$. As $\omega'$ achieves the same minimum mean squared error on the $k$-th environment, $\omega'$ must satisfy the normal equation, too. Thus, no spurious information will be included and $X\Phi = [SX_v; \mathbf{0}_{d_s}]$ where $S \in \mathbb{R}^{d_v \times d_v}$ is an invertible matrix and $\mathbf{0}_{d_s}$ denotes a $d_s$-dim vector of all zeros.

**Step 2:** No invariant information will be discarded. Under Condition 1 where $H(\cdot|\cdot)$ is replaced with $L(\cdot|\cdot)$, we have $L(Y|X_v) - L(Y|X_v, \rho(Z)) = 0$. Then $X\Phi$ will satisfy the constraint of Problem 8. Notice that $X\Phi$ achieves the smallest loss when only using invariant feature information.

Combining these two steps completes the proof. $\qquad\square$

### A.3 More Discussions on Assumption 2

This assumption aims to ensure that the invariance penalty cannot be arbitrarily small if a spurious feature, together with other features, is selected by a feature mask. For example, in the extreme case where $X_2$ is the only invariant feature (i.e., $X_v$ consists of only $X_2$) and can perfectly predict $Y$, we would have $H(Y|X_2) = 0$. Then for a spurious feature $X_1$, $H(Y|X_1, X_2) = 0$ and $H(Y|X_1, X_2, \rho(Z)) = 0$ for any $\rho(\cdot)$, and we cannot identify $X_1$ as the spurious feature. Nevertheless, since there are generally exogenous noise variables in the SCM and $H(Y|X_v)$ is positive, we believe that this assumption holds in most cases.

## B Proofs

### B.1 Proof of Theorem 1

*Proof.* First, we assume $Y$ and $X_v$ are univariate variables, i.e., $Y, X_v \in \mathbb{R}$. Let $\eta_1 \sim \text{Uniform}(0, 1)$ independent of $X_s$ and $Y$, and set $X'_v = X_s$. Define the conditional cumulative distribution function and its inverse as:

$$F_{Y|X_s = x_s}(y) = P(Y \le y \mid X_s = x_s),$$
$$Y' = f'_1(X'_v, \eta_1) = F_{Y|X_s}^{-1}(\eta_1) = \inf\{y \in \mathbb{R} : F_{Y|X_s}(y) \ge \eta_1\} \text{ with } X_s = X'_v.$$

By definition, we would have

$$
\begin{aligned}
P(Y' \le y \mid X'_v) &= P(f'_1(X'_v, \eta_1) \le y) \\
&= P(F_{Y|X_s}^{-1}(\eta_1) \le y) \\
&= P(F_{Y|X_s} \circ F_{Y|X_s}^{-1}(\eta_1) \le F_{Y|X_s}(y)) \\
&= P(\eta_1 \le F_{Y|X_s}(y)) \\
&= P(Y \le y \mid X_s).
\end{aligned}
\tag{9}
$$

Similarly, we can construct $X'_s = f'_2(X'_v, Y', \eta_2) = F_{X_v|Y, X_s}^{-1}(\eta_2)$ so that $P(X'_s \mid Y', X'_v) = P(X_v \mid Y, X_s)$ with $\eta_2 \sim \text{Uniform}(0, 1)$. Thus, we have

$$
\begin{aligned}
P(X'_v, X'_s, Y) &= P(X'_s \mid Y', X'_v) P(Y' \mid X'_v) P(X'_v) \\
&= P(X_v \mid Y, X_s) P(Y \mid X_s) P(X_s) \\
&= P(X_v, X_s, Y).
\end{aligned}
\tag{10}
$$

Next, we can easily find a function $q'(\cdot)$ so that $X' = q'(X'_v, X'_s) = q(X_v, X_s)$, where we have chosen $X'_v = X_s$ and $X'_s = X_v$. Together with Equation (10), we conclude that $P(X', Y') = P(X, Y)$.

Second, we consider that $\boldsymbol{X}_v$ has a multi-dimension. We may leave $Y$ to be a univariate variable as the label in many ML problems is scalar-valued. For this case, we can pick an entry of $\boldsymbol{X}_v$, say, $\boldsymbol{X}_v^{(j)}$ for some $j$. Then we set $\boldsymbol{X}_v' = (\boldsymbol{X}_v^{(-j)}, \boldsymbol{X}_s)$ where $\boldsymbol{X}_v^{(-j)}$ denotes the rest entries of $\boldsymbol{X}_v$ except the $j$-th. Then we can similarly use the inverse conditional cumulative distribution function and an independent uniformly distributed noise variable $\eta_1$ to construct $Y' = f'(\boldsymbol{X}_v', \eta_1)$ so that $P(Y' \leq y \mid \boldsymbol{X}_v') = P(Y \leq y \mid \boldsymbol{X}_v^{(-j)}, \boldsymbol{X}_s)$. Similarly, we set $\boldsymbol{X}_s' = \boldsymbol{X}_v^{(j)}$ and we can construct $\boldsymbol{X}_s' = f_2'(\boldsymbol{X}_v', Y', \eta_2)$ so that $P(\boldsymbol{X}_s'|Y', \boldsymbol{X}_v') = P(\boldsymbol{X}_v^{(j)} \mid Y, \boldsymbol{X}_s)$. We can then get the same conclusion following the previous proof. □

### B.2 Proof of Theorem 2

*Proof.* Our proof proceeds by two steps. First, we show that any feature mask that selects at least one spurious feature would induce a penalty. With sufficiently large $\lambda$, the penalty will dominate the expected risk and then exceed $\hat{\mathcal{L}}(\Phi_v)$. Second, we show that any proper subset of the invariant features induces a loss larger than $\hat{\mathcal{L}}(\Phi_v)$.

**Step 1** Suppose that the feature mask contains at least one spurious features. Denote the selected features as $\boldsymbol{X}_{+s}$ and the corresponding feature mask as $\Phi_{+s}$. We aim to show that

$$\hat{\mathcal{L}}_{(\Phi_{+s})} > \hat{\mathcal{L}}(\Phi_v).$$

By Assumption 1 with a given $\epsilon > 0$, we have

$$
\begin{aligned}
\hat{\mathcal{L}}(\Phi_v) &\leq (1+2\lambda)\epsilon + H(Y|\boldsymbol{X}_v) + \lambda\left(H(Y|\boldsymbol{X}_v) - H(Y|\boldsymbol{X}_v, \rho(Z))\right) \\
&= (1+2\lambda)\epsilon + H(Y|\boldsymbol{X}_v) \\
&\leq (1+2\lambda)\epsilon + H(Y).
\end{aligned}
\tag{11}
$$

One the other hand, we have

$$
\begin{aligned}
\mathcal{L}(\Phi_{+s}) &\geq -(1+2\lambda)\epsilon + H(Y|\boldsymbol{X}_{+s}) + \lambda\left(H(Y|\boldsymbol{X}_{+s}) - H(Y|\boldsymbol{X}_{+s}, \rho(Z))\right) \\
&\geq -(1+2\lambda)\epsilon + \lambda\left(H(Y|\boldsymbol{X}_{+s}) - H(Y|\boldsymbol{X}_{+s}, \rho(\boldsymbol{Z}))\right) \\
&\geq -(1+2\lambda)\epsilon + \lambda\delta C,
\end{aligned}
\tag{12}
$$

where the last inequality is due to Assumption 2 and Condition 2. Thus, if we choose $\epsilon < \delta C/4$ and $\lambda > \frac{H(Y)+2\epsilon}{\delta C - 4\epsilon}$, we can get

$$\mathcal{L}(\Phi_v) < \mathcal{L}(\Phi_{+s}).$$

**Step 2** Denote a proper subset of invariant features as $\boldsymbol{X}_{-v} \subsetneq \boldsymbol{X}_v$, and similarly the feature mask as $\Phi_{-v}$.

In Step 1, we have shown that

$$\hat{\mathcal{L}}(\Phi_v) \leq (1+2\lambda)\epsilon + H(Y|\boldsymbol{X}_v).$$

Similar to Equation (12), we have

$$\hat{\mathcal{L}}(\Phi_{-v}) \geq -(1+2\lambda)\epsilon + H(Y|\boldsymbol{X}_{-v}).$$

Then according to Assumption 3, we have

$$
\begin{aligned}
\hat{\mathcal{L}}(\Phi_{-v}) - \hat{\mathcal{L}}(\Phi_v) &\geq -2(1+2\lambda)\epsilon + H(y|\boldsymbol{X}_{-v}) - H(y|\boldsymbol{X}_v) \\
&\geq -2(1+2\lambda)\epsilon + \gamma.
\end{aligned}
$$

Thus, if $\epsilon < \frac{\gamma}{2(1+2\lambda)}$, we have

$$\hat{\mathcal{L}}(\Phi_{-v}) > \hat{\mathcal{L}}(\Phi_v).$$

In conclusion, with $\lambda \in [\frac{H(Y)+1/2\delta C}{\delta^d C - 4\epsilon} - \frac{1}{2}, \frac{\gamma}{4\epsilon} - \frac{1}{2}]$, we can get

$$\hat{\mathcal{L}}(\Phi_v) < \hat{\mathcal{L}}(\Phi), \quad \forall\Phi \neq \Phi_v.$$

Notably, there exists a feasible $\lambda$ if $\epsilon < \frac{C\gamma\delta}{4\gamma + 2C\delta H(Y)}$. The proof is complete by noticing that $\epsilon$ can be chosen arbitrarily according to Assumption 1. □

## B.3 Proof of Proposition 1

*Proof.* Consider the following feature set

$$\boldsymbol{X}_{\bar{v}} := \max_{|\boldsymbol{X}'|}\{\boldsymbol{X}' \subset \boldsymbol{X} : H(Y|\boldsymbol{X}, \rho(\boldsymbol{Z})) = H(Y|\boldsymbol{X}) \quad \forall \rho(\cdot)\},$$

and the corresponding feature mask is denoted as $\Phi_{\bar{v}}$. It corresponds to the largest subset of $\boldsymbol{X}$ that satisfies the invariance constraint.

Note that $\Phi_{\bar{v}} \neq \Phi_v$. By Assumption 1, for a given $\epsilon$ we can get

$$
\begin{aligned}
\hat{\mathcal{L}}(\Phi_{\bar{v}}) &\leq (1 + 2\lambda)\epsilon + H(Y|\boldsymbol{X}_{\bar{v}}) + \lambda(H(Y|\boldsymbol{X}_{\bar{v}}) - H(Y|\boldsymbol{X}_v, \rho(\boldsymbol{Z}))) \\
&= (1 + 2\lambda)\epsilon + H(Y|\boldsymbol{X}_{\bar{v}}) \\
&\leq (1 + 2\lambda)\epsilon + H(Y).
\end{aligned}
$$

One the other hand, we have

$$
\begin{aligned}
\mathcal{L}(\Phi_v) &\geq -(1 + 2\lambda)\epsilon + H(Y|\boldsymbol{X}_v) + \lambda\left(H(Y|\boldsymbol{X}_v) - H(Y|\boldsymbol{X}_v, \rho(\boldsymbol{Z}))\right) \\
&\geq -(1 + 2\lambda)\epsilon + \lambda\left(H(Y|\boldsymbol{X}_v) - H(Y|\boldsymbol{X}_v, \rho(\boldsymbol{Z}))\right) \\
&\geq -(1 + 2\lambda)\epsilon + \lambda\delta C',
\end{aligned}
$$

which can be shown similarly to Equation (12). Thus, if we choose $\epsilon < \delta C'/4$ and $\lambda > \frac{H(Y)+2\epsilon}{\delta C'-4\epsilon}$, we would get

$$\mathcal{L}(\Phi_{\bar{v}}) < \mathcal{L}(\Phi_v).$$

$\square$

## B.4 Proof of Corollary 1

*Proof.* (a) Since $h$ is injective, $H(Y|\boldsymbol{X}_v, h(Y)) = H(Y|\boldsymbol{X}_v, Y) = 0$ for any $\boldsymbol{X}_v$. By Assumption 3, we have $H(Y|\boldsymbol{X}_v) \geq H(Y|\boldsymbol{X}) + \gamma \geq \gamma$. Then Proposition 1 indicates that we cannot identify all the invariant features. (b) The proof proceeds the same as above by noting $H(Y|\boldsymbol{X}_v, h(\boldsymbol{X}, Y)) = 0$. (c) This case can be shown similarly to (a), because $H(Y|X_v, h(\text{Index}(\boldsymbol{X}, Y))) = H(Y|X_v, \boldsymbol{X}, Y) = 0$. $\square$

## B.5 Proof of Proposition 2

*Proof.* Denote a feature set $\boldsymbol{X}_{v(+k)}$, which contains the invariant feature set $\boldsymbol{X}_v$ as well a spurious feature $\boldsymbol{X}_s^k$ in Proposition 2.

Similar to Eq. (11), we can show that

$$\hat{\mathcal{L}}(\Phi_{v(+k)}) \leq (1 + 2\lambda)\epsilon + H(Y|\boldsymbol{X}_{v(+k)}).$$

And similar to Eq. (12), we also have

$$\hat{\mathcal{L}}(\Phi_v) \geq -(1 + 2\lambda)\epsilon + H(Y|\boldsymbol{X}_v).$$

Then it follows that

$$
\begin{aligned}
\hat{\mathcal{L}}(\Phi_v) - \hat{\mathcal{L}}(\Phi_{v(+k)}) &\geq -2(1 + 2\lambda)\epsilon + H(Y|\boldsymbol{X}_v) - H(y|\boldsymbol{X}_{v(+k)}) \\
&\geq -2(1 + 2\lambda)\epsilon + \gamma.
\end{aligned}
$$

If $\epsilon < \frac{\gamma}{2(1+2\lambda)}$, we have

$$\hat{\mathcal{L}}(\Phi_v) > \hat{\mathcal{L}}(\Phi_{v(+k)}).$$

Thus, we cannot identify all the invariant features from Problem 6. $\square$

### B.6 Proof of Meeting Condition 1

We show that $H(Y|\boldsymbol{X}_v, \rho(\boldsymbol{Z})) = H(Y|\boldsymbol{X}_v)$ for all $\rho(\cdot)$ if $H(Y|\boldsymbol{X}_v, \boldsymbol{Z}) = H(Y|\boldsymbol{X}_v)$ holds.

*Proof.* On one hand, because $\rho(\boldsymbol{Z})$ contains less information than $\boldsymbol{Z}$, we have

$$H(Y|\boldsymbol{X}_v, \rho(\boldsymbol{Z})) \geq H(Y|\boldsymbol{X}_v, \boldsymbol{Z}) = H(Y|\boldsymbol{X}_v).$$

On the other hand, $\boldsymbol{X}_v$ and $\rho(\boldsymbol{Z})$ contain more information than $\boldsymbol{X}_v$, so we can get

$$H(Y|\boldsymbol{X}_v, \rho(\boldsymbol{Z})) \leq H(Y|\boldsymbol{X}_v).$$

Thus, we conclude $H(Y|\boldsymbol{X}_v, \rho(\boldsymbol{Z})) = H(Y|\boldsymbol{X}_v)$. □

## C  More Experimental Details

**Implementing Approximated ZIN**. Since Eq. (5) is a challenging minimax formulation, we replace the penalty term given a environment partition with its first order approximation. Specifically, we consider the following surrogate minimax formulation:

$$\min_{\omega, \Phi} \max_{\rho} = \mathcal{R}(\omega, \Phi) + \lambda \sum_{k=1}^{K} \|\nabla_\omega \mathcal{R}_{\rho^{(k)}}(\omega, \Phi)\|^2. \tag{13}$$

Readers can refer to the Appendix B.3 of [57] for more details about the relationship between Eqs. (5) and (13).

We further use a two-stage method to approximate the minimax procedure:

- Minimize $\mathcal{R}(\omega, \Phi)$ over $(\omega, \Phi)$ and simultaneously maximize $\sum_{k=1}^{K} \|\nabla_\omega \mathcal{R}_{\rho^{(k)}}(\omega, \Phi)\|^2$ over $\rho$.
- Fix $\rho$ and minimize Eq. (13) over $(\omega, \Phi)$.

**Training Details**. For the synthetic datasets and the house price prediction dataset, we use the full batch gradient in optimization. The ERM method is trained for 4000 epochs. The IRM/ZIN method is also trained for 4000 epochs, with additional annealing in the first 2000 epochs. We train EIIL for 4000 epochs which is divided equally, i.e., 2000 epochs, in each of the two stages. For the CelebA classification task, we use mini-batch training with batch size of 128. All the methods are trained for 50 epochs, with annealing strategy in the first 25 epochs. For the Landcover task, we use mini-batch training with batch size of 1024, and all the methods are trained for 400 epochs with annealing strategy in the first 40 epochs. We use Adam [23] with learning rate 0.001 as our optimizer. Our experiments are run in a Linux workstation with Intel Xeon 3.20GHz CPU, 128GB RAM, and Nvidia GTX 3090 GPU. It takes about 5 GPU hours for the CelebA tasks, while the house price task and Landcover task cost less than 10 minutes for training. Notably, since IRM suffers from overfitting problem when applied to large models like ResNet-18 [27, 57], we fix the feature extraction backbone (that is, use a pre-trained model) in the CelebA experiment to allieviate this issue.

## D  More Experiment Results

### D.1 Spatial Heterogeneity

We also consider spatial heterogeneity that is also commonly encountered in practice, e.g., environments may be divided according to locations of latitude and longitude. We simulate spatial heterogeneity in the same way the data generation process for time heterogeneity but use a two-dimensional spatial variable $\boldsymbol{r} = [r_1, r_2] \in [0, 1]^2$. We simulate four environments for training by equally splitting the space into four blocks, i.e., $\{[0, 0.5) \times [0, 0.5), [0, 0.5) \times [0.5, 1], [0.5, 1] \times [0, 0.5), [0.5, 1] \times [0.5, 1]\}$. Similarly, we denote a simulated case by tuple of $p_s(\boldsymbol{r})$ in the four elements. We also evaluate the performance on four distinct test environments with $p_s \in \{0.999, 0.8, 0.2, 0.1\}$ and $p_v$ being constant. More details regarding the implementations are given in Appendix C.

The experimental results of spatial heterogeneity in Table 6 show similar performances to those of temporal heterogeneity. An interesting observation is that ZIN outperforms IRM with ground-truth environments in this simulation, especially when "duplicated" environments exist, e.g., when

**Table 6:** Test Mean and Worst accuracy (%) on four spatial heterogeneity synthetic datasets.

| Env Partition | $p_s(t)$ | 0.999, 0.999, 0.7, 0.7 | | | | 0.999, 0.9, 0.8, 0.7 | | | | 0.999, 0.999, 0.8, 0.8 | | | |
|---|---|---|---|---|---|---|---|---|---|---|---|---|---|
| | $p_v$ | 0.9 | | 0.8 | | 0.9 | | 0.8 | | 0.9 | | 0.8 | |
| | Test Acc | Mean | Worst | Mean | Worst | Mean | Worst | Mean | Worst | Mean | Worst | Mean | Worst |
| No | ERM | 76.65 | 59.48 | 60.33 | 27.25 | 76.59 | 59.35 | 60.30 | 27.25 | 69.93 | 44.65 | 56.23 | 16.60 |
| | EIIL | 37.81 | 16.89 | 66.46 | 50.35 | 37.03 | 14.01 | 66.60 | 50.28 | 70.18 | 45.23 | 71.00 | 58.72 |
| | HRM | 49.98 | 49.95 | 49.97 | 49.92 | 49.99 | 49.98 | 50.00 | 49.99 | 49.97 | 49.95 | 49.99 | 49.97 |
| | ZIN | **88.66** | **87.23** | **79.16** | **78.04** | **88.28** | **86.29** | **78.92** | **77.49** | **88.00** | **85.75** | **78.80** | **77.25** |
| Yes | IRM | 83.71 | 82.24 | 73.26 | 71.25 | 86.73 | 83.79 | 75.80 | 73.33 | 84.39 | 81.48 | 73.15 | 69.97 |

$p_s(\boldsymbol{r}) = (0.999, 0.999, 0.7, 0.7)$. We conjecture that in this case the "ground-truth" partition may not be the most effective for invariant learning due to the heavily overlapped environments. As shown in the right panel of Figure 2, ZIN automatically merges duplicated environments.

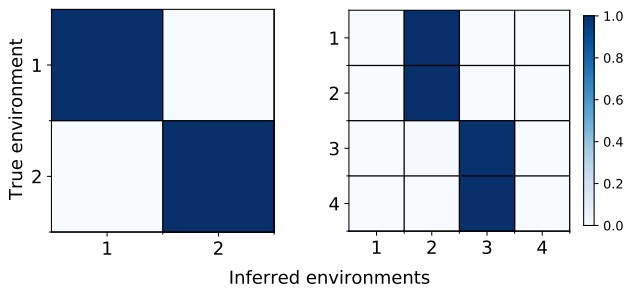

**Figure 2:** Visualization of inferred environments. **Left**: temporal heterogeneity setting $(0.999, 0.7)$. **Right**: spatial heterogeneity setting $(0.999, 0.999, 0.7, 0.7)$.

### D.2 Ablation Study on Auxiliary Information and Number of Environments

We now verify the theoretical results in Section 5 by choosing different inputs as $\boldsymbol{Z}$. We adopt a setting of spatial heterogeneity with $p_s(\boldsymbol{r}) = (0.999, 0.999, 0.8, 0.8)$ and $p_v = 0.8$. Note that the heterogeneity in this setting is only along the second dimension $r_2$. The results are shown in Table 7. One can verify that Conditions 1 and 2 are satisfied when $\boldsymbol{r}$ or $r_2$ is chosen as $\boldsymbol{Z}$. The mean and worst test accuracy implies that ZIN based on $\boldsymbol{r}$ or $r_2$ can effectively remove the spurious feature. Since $p_s(\boldsymbol{r})$ does not change along $r_1$, choosing $r_1$ to infer environment partition violates Condition 2, which is reflected by the poor performance of ZIN with $r_1$. Using $[\boldsymbol{X}, Y]$ as input to $\rho(\cdot)$ is also a violation by Corollary 1, and the corresponding results in Table 7 confirm our analysis. Lastly, notice that $X_s \in \boldsymbol{X}$ contains some information of $Y$ and there may exist a function $\rho(\cdot)$ so that $H(Y|X_v) > H(Y|X_v, \rho(\boldsymbol{X}))$. This violates Condition 1 and leads to poor results when choosing $\boldsymbol{X}$ as input to $\rho(\cdot)$.

**Table 7:** Ablation study on choice of $\boldsymbol{Z}$.

| $\boldsymbol{Z}$ | Condition 1 | Condition2 | Test Mean | Test Worst |
|---|---|---|---|---|
| $\boldsymbol{r}$ | ✓ | ✓ | 78.80 | 77.25 |
| $r_1$ | ✓ | ✗ | 56.30 | 16.84 |
| $r_2$ | ✓ | ✓ | 78.79 | 77.21 |
| $\boldsymbol{X}$ | ✗ | ✓ | 59.85 | 25.99 |
| $\boldsymbol{X}, Y$ | ✗ | ✓ | 71.09 | 58.85 |

We also empirically verify the choice of hyper-parameter $K$ using the same simulated setting. Fig. 3 shows that $K$ has a relatively small impact, especially when $K \geq 4$.

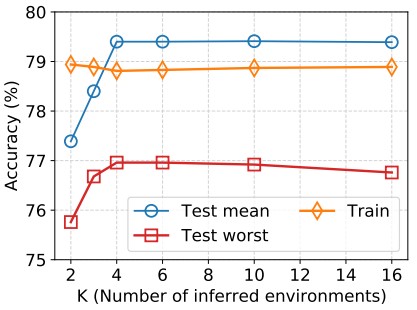 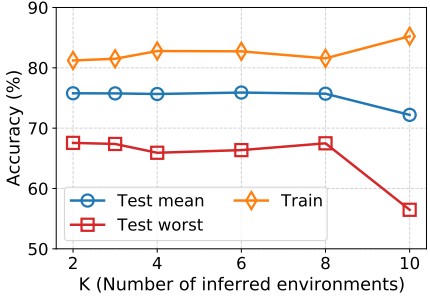

**Figure 3:** Ablation study on the choice of $K$. **Left**: synthetic dataset. **Right**: CelebA.

### D.3 About the Inferred Environments on CelebA

We visualize the inferred environments of the CelebA dataset in this section. The target *Smiling* is spuriously correlated with feature *Gender*, i.e., most females are smiling while most males are not. We set $K$ to be 2 (our framework is insensitive to $K$ as shown in the right panel of Fig. 3). We visualize the spurious correlations in the two inferred environments during training in Fig. 4. ZIN can generate two environments where the spurious correlations differ. Then we can easily discard the spurious feature using IRM methods on the inferred environments.

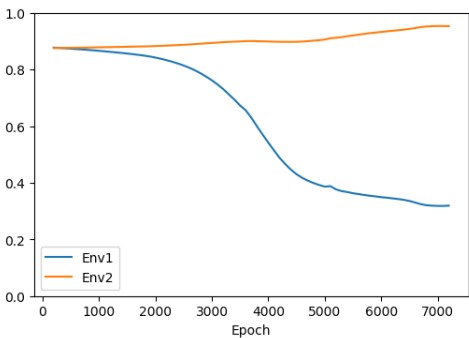

**Figure 4:** Correlation of the spurious feature (Gender) with the target (Smiling). The spurious correlation is calculated as the percentage of samples whose target (Smiling/Not Smiling) aligns with its gender (Female/Male). For example, a smiling female or a non-smiling male is counted as score 1, otherwise as score 0. The average score represents the correlation between Smiling and Gender.

## E Examples of Valid and Invalid Choices of Auxiliary Information

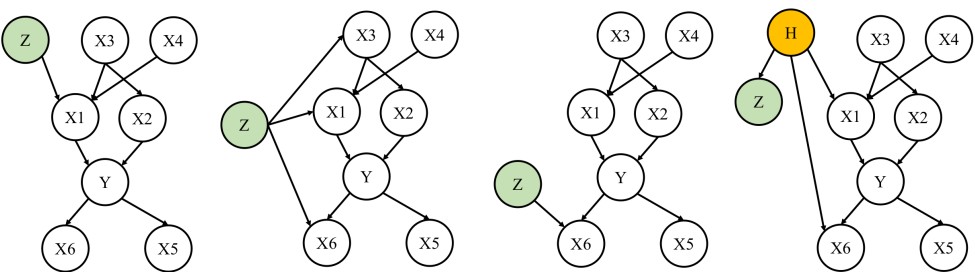

**Figure 5:** Examples of valid choices of $Z$ satisfying Condition 1. Here "H" in the 4th graph denotes some hidden confounders. The invariant features are $X_1$ and $X_2$, direct causes of $Y$.

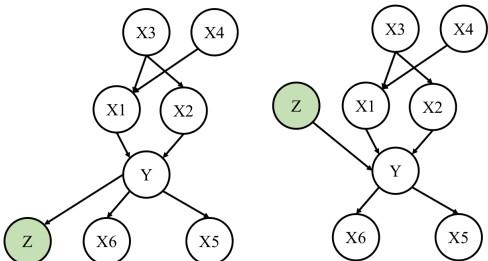

**Figure 6:** Invalid choices of $Z$ that violate Condition 1. The invariant features are $X_1$ and $X_2$, the direct causes of $Y$.

# F  Societal Impact

In this work, we propose to utilize the auxiliary information to aid the invariance learning without environmental indexes. This method is also helpful to fairness issues; see, e.g., the discussion about out-of-distribution generation and algorithmic fairness in [10]. For the additional information, we should also avoid using demographic, private, and sensitive information.