# OpenReview forum: "ZIN: When and How to Learn Invariance Without Environment Partition?"
_NeurIPS.cc/2022/Conference — NeurIPS 2022 Accept_

### Official Review · Reviewer_TsN9 · 2022-07-12

**Rating:** 8
**Confidence:** 4
**Soundness:** 4 excellent
**Presentation:** 4 excellent
**Contribution:** 3 good

**Summary:**

There is a recent line of work that build on methods that learn invariant representations from multiple environments [Peters et al 2016, Arjovsky et al 2019] by attempting to infer the environment label from data. This paper presents a simple counterexample that shows that environment inference is not possible in general---essentially they construct examples where the joint is consistent with two different SCMs, each of which have a different invariant feature; and hence any environment inference procedure that depended only on the joint distribution would be wrong in at least one of the SCMs---and instead argues for environment inference from auxiliary variables. They complement their negative results with theory that shows that auxiliary variables are sufficient for environment inference in the linear and feature selection settings, and give nice experimental results on both simulated and real data.

**Questions:**

1. Why is assumption 2 necessary? If you weren't making it explicitly, I would have thought it was just a consequence of your definition of invariance (line 116 / 117). What is it ruling out?

2. Related to my weakness comment - many of the selected auxiliary variables seem very similar to an adjustment set that you would control for in order to block paths to the environment. Is there an explicit connection here? Can we interpret the method as an invariant representation learning alternative to adjustment? And if so, does ZIN fail in the presence of unblocked confounding?

**Limitations:**

Yes

**Strengths And Weaknesses:**

This review is going to skew short because I really liked this paper: I've long thought that environment inference was asking too much so I really appreciate a clear counter example that shows where it can fail (of course, that still leaves open the question of sufficient conditions for env inference to succeed without auxiliary info; but that's a separate question), and the auxiliary information perspective is a nice way around the negative examples. I also appreciated that the empirical section focused on more realistic datasets.

Weaknesses:
* The biggest weakest for me was that it doesn't seem obvious how one should choose an auxiliary variable in practice. There are a nice collection of examples in the experimental section which partially addresses this, but I think it could have been strengthened by including a section where you discuss the process by which you selected the auxiliary variables. For each of the datasets, you could explain what variables are available, why the selected subset make sense as auxiliary variables, and under what conditions you'd expect environment inference to fail.

---

> ### Author Response · Authors · 2022-08-02
> **Response**
>
> ## Weakness
>
> ### Q1 How to find $Z$ in practice.
>
> Thank you for the insightful question and we indeed miss this part in our original submission. In the revised version, we add a section (Appendix F) on the causal interpretations of $Z$ with specific causal graphs and examples. We will try to fit this part into the main paper in the future version because it is of vital importance.
> We briefly summarize the results in Appendix F.
> * **How to satisfy Condition 1**. The path between  $Z$ and $Y$ should be $D$-separated by $X_v$. Notably, $Z$ can’t be the parent or the child of $Y$ as shown in Figure 6. Figure 4 also shows a concrete example of the meta information in the image classification tasks together with a illustrating causal graph. The meta information of a image, e.g., the time slot, coordinate and temperate, does not have a direct effect on the target.  It has some correlations with some nodes of the causal system.  Then we have $H(Y|X_v)=H(Y|X_v, Z)$. This meta information serves as valid $Z$. Figure 5 illustrates more valid choices of $Z$ with causal explanations.
> * **How to satisfy Condition 2**. Condition 2 is hard to check in practice. While $Z$ cannot be the direct parent or the child of $Y$ (otherwise violating Condition 1), $Z$ should be correlated with some nodes in the interested causal graph. Further, we show an additional theoretical result in Appendix F.2: even if Condition 2 is only partially satisfied, we can still discard the spurious features that are distinguishable by the collected $Z$. At the same time, the invariant features are preserved. As we collect more $Z$ satisfying Condition 1, we can discard more spurious features.
>
> In conclusion, we should try to **find as many $Z$, which satisfies Condition 1 and is not independent of the nodes in the causal graph, as possible**.
>
> **Please refer to Appendix F for the detailed contents**.
> Thanks again for the constructive suggestions.
>
> ## Questions
>
> ### Q1 Why we make Assumption 2.
>
> Assumption 2 is  made mainly due to technical concerns. It is *only* used in the proof of Proposition 2 in Section 5.3. Proposition 2 says if there is a spurious feature $X_s^k$ that looks stable in the environments, then a feature mask $\Phi_{v+k}$ selecting this spurious feature $X_s^k$ and all invariant features $X_v$ will achieves smaller risk than ideal feature mask $\Phi_{v}$.  In the proof, we need to show $\Phi_{v+k}$ will not induce a penalty because both $X_k$ and $X_v$ look ‘invariant’.
>
> We have also been thinking about dropping this assumption. Actually we did not find any failure cases of Assumption 2. However, we also find it  difficult to provide rigorous and general justification for  Assumption 2. So we will keep working on this question and hopefully figure out how to relax it.
>
> ### Q2 Regarding adjustment set and confounding bias.
>
> This is a great question. Indeed, we did not think about the role of auxiliary information in this way, but rather place some conditions (conditions 1 and 2) on the auxiliary information $Z$. Perhaps the very first thing is to introduce the environment in the causal graph or the SCM. A possible way is to put it in structural equation of $X_s$. Intuitively, we need auxiliary information to have some information about the environment, and $Z$ can be a child of the environment in this sense. So $Z$ may not necessarily be the adjustment variable that blocks the paths into environment.
>
> We have to say that we cannot provide a firm conclusion at present. We again greatly appreciate the reviewer’s insightful question and will continue this direction to come with sufficient conditions for selecting the auxiliary information.

---

> > ### Comment · Reviewer_TsN9 · 2022-08-07
> > **Thanks**
> >
> > Thanks for the responses and congratulations on a nice paper.

---

### Official Review · Reviewer_9XzW · 2022-07-14

**Rating:** 8
**Confidence:** 3
**Soundness:** 4 excellent
**Presentation:** 4 excellent
**Contribution:** 3 good

**Summary:**

This work explores the feasibility of utilizing cheaply available auxiliary features to uncover latent environment partition and help to train an invariant model. The proposed method, namely ZIN, is a min-max game. Specifically, the learned environment partition aims to split samples into groups with different distributions by maximizing the performance gap between environments. The feature extractor and the classifier aim to learn causally invariant features by minimizing such performance gaps. The proposed model is significant when ground environment labels are costly to obtain while additional information that correlates with environments is easy to acquire.

**Questions:**

1. According to the author, it is theoretically impossible to learn invariant features from heterogeneous data without environment indexes. A counter-example and a general theoretical proof are provided by the author.
a). For counter-example, two examples are symmetric. In other words, the causal graph cannot be identified given the data. Both X1 and X2 can be treated as either invariant features or spurious features. It is not indistinguishable, but more like both are right. Additionally, the data generation process is impractical. It is not a reasonable example, more like a math game.
b). For the general theoretical result, the proof is under the specific SCM (fully informative invariant features). However, there are other widely used assumptions [1]. Personally, I don't think the timing is ripe to come to such a firm conclusion when a theoretical proof under only one specific assumption is proposed.

2. More information and discussion can be proposed for the methods section. Why the invariance penalty is effective? Why are different classifiers being used for various estimated environments?

3. Only IRM, used as an Oracle baseline, has been compared in the experiments. What about a different class of generalization-enhancing techniques, like group DRO?

4. Figure 1 shows the correlation between inferred environments and true environments in synthetic data, which is important. Also, Figure 2 demonstrates the effect of K - the number of inferred environments, which is crucial for ZIN. Since K, as a hyperparameter, is pre-specified in ZIN. Then how about the situation in real data? Visualization of the environment and the impact of K on real data is essential to demonstrating the efficacy of ZIN. Could you provide, for instance, the distribution of two environments for gender in CelebA?

A small question: It seems that the additional auxiliary variables should have a strong correlation with spurious variables, so that the distribution of spurious variables is distinct in different environments. And the classifier can further remove the dependency. For the regression dataset of house sales price, why the built year is auxiliary information for ZIN? I think it is a cause for predicting the price.



[1] Invariance Principle Meets Information Bottleneck for Out-of-Distribution Generalization.



Post rebuttal comment
---------------------------------------------------
I thank the authors for providing their feedback and addressing all my concerns.

I was already convinced of the quality of the paper. Reading comments by the authors did nothing other than reinforce my assessment. After the rebuttal, I would like to raise my score from 6 to 8.

**Ethics Review Area:**

["I don’t know"]

**Limitations:**

There is no negative societal impact of this work.

**Strengths And Weaknesses:**

From a personal perspective, the key strength of this paper is its training of an end-to-end model with additional auxiliary information to address environment inference. This idea, for me, is novel and interesting.

However, I still have some concerns about the content of this paper.

1. The author makes a very strong conclusion - without inductive biases or additional information, environment inference is impossible, but neither the example nor the theorem convinced me. Does it contest the theoretical viability of resampling-based techniques like LrF?

2. The Methodology section's structure could be enhanced.

3. Some important baselines, such as "Group DRO" are missing.

4.Important case studies are missing - including the visualization of environments and the effect of K on real data.

The following part has more information.

---

> ### Author Response · Authors · 2022-08-02
> **Response 2/2**
>
> ## Q2. Response to "The Methodology section’s structure could be enhanced."
>
> We have reformulated our methodology section in the revised version according to your advice. Specifically, we introduce the involved variables and terms before presenting the formulation. We also first introduce the invariance penalty before the joint environmental inference framework. See Section 5.1 (marked in blue) for details.
>
> Suppose that the environments $(1,...,k,...,K)$ have been given according to a fixed $\rho(\cdot)$. Recall that IRM [1] learns an invariant representation, upon which there is a classifier that is simultaneously optimal in all environments. To measure the optimality of a classifier $\omega$ in environment $k$, we can fit an environment dependent classifier $\omega_k$ on the data from environment $k$. If $\omega_k$ achieves a smaller loss than $\omega$, it means $\omega$ is not optimal in this environment. We can further train a set of environment dependent classifiers $\\{\omega_k\\}_{k=1}^K$, to measure whether $\omega$ is simultaneously optimal in all environments.
> Then we search for environment partition that induces the maximum invariance penalty. Hopefully, this kind of environments can help us to distinguish between spurious and invariant features.
>
> [1] Invariant Risk Minimization
>
> ## Q3 Additional experimental results of GroupDRO on House Price and CelebA
>
> We have added GroupDRO for the  House Price and CelebA experiments, where we assume the underlying environments are known, similar to IRM. The results are included in Tables 2 and 3. We observe that GroupDRO is inferior to IRM (with environment indexes) and ZIN (without environment indexes).
>
> ## Q4 The hyperparamter K on real data (CelebA). The distribution in the two environments of CelebA.
>
> We conduct experiments of different $K$ (K=2,3,4,6,8,10) on CelebA and add more results in Appendix E.2. As Figure 2 (Right) shows, the results are quite stable when we change K between 2-8.
>
> Furthermore, we visualize the distributions of the spurious feature in CelebA in the two environments (K=2) in Appendix E.3. Specifically, we calculate the spurious correlation  as the percentage of samples whose target (Smiling/Not Smiling) aligns with its gender (Female/Male). The results are shown in Figure 3 of Appendix E.3. We can see the spurious correlation differs greatly in the learned environments at the end of  training. This means that ZIN can generate environments among which the spurious feature exhibits non-invariance and further we can apply IRM to learn the invariant features.
>
> ## Q5. For the regression task, why the built year is auxiliary information for ZIN?
>
> This is a good point. Typically, the built year is a cause of the price. Here our task was to predict the *ranking* of the house price in the same built year. Thus, the prices of houses with the same built year are normalized, and in this way, the built year is no longer a cause of the target. We forgot to mention this task explicitly and have added a description in the revision. Again we appreciate the reviewer's effort.

---

> > ### Comment · Reviewer_9XzW · 2022-08-04
> > **Post rebuttal comment**
> >
> > I thank the authors for providing their feedback and addressing all my concerns.
> >
> > I was already convinced of the quality of the paper. Reading comments by the authors did nothing other than reinforce my assessment. After the rebuttal, I would like to raise my score from 6 to 8.

---

> ### Author Response · Authors · 2022-08-02
> **Response 1/2**
>
>
> ## Q1. Response to ''The author makes a very strong ... resampling-based techniques like LrF''.
>
> ###  Q1-1. For the implication of Example 1. Whether the resampling-based techniques like LrF can work.
>
> The counterexample shows that when we observe a joint distribution $P(X_1, X_2, Y)$, there may exist two possible SCMs governing the data generation process. In the first SCM, $X_1$ is invariant and $X_2$ is spurious. In the second, $X_1$ is spurious. There is only one true SCM but we cannot  know which one from the joint distribution. Notice that the underlying SCM determines the desired invariance, and in fact the two processes can generate very different test distributions (depending on what is the spurious feature). Indeed, our impossibility result for this setting is in the same spirit with the identifiability issue in causal discovery and ICA, where one has to introduce additional assumptions/conditions to have a meaningful problem.
>
> Concretely, consider that a deterministic algorithm $A$ is applied to the joint distribution (mixed dataset). It would output some invariant features. WLOG, assume that $A(P(X_1, X_2, Y)) = X_1$. Then  $A$ will also return $X_1$ as the invariant feature in the second process because it induces exactly the same joint distribution. Since $X_1$ is the spurious feature in the second case, algorithm $A$ relies on $X_1$ and would fail in the testing distribution where $X_1$ can change dramatically.  One can also check when $A$ deterministically outputs $X_2$ or both,  or $A$ is a randomized algorithm.
>
> Does the method “LrF” refer to  “Learning from Failure” [1]? If so, we think that this method also fails in the above example, because it would select the same "failure" samples and will deterministically some feature as invariant feature depending on its inductive bias.
> (We also discuss this problem further in the next sub question).
>
> [1] Learning from Failure: Training Debiased Classifier from Biased Classifier
>
> ### Q1-2. The data generation process of the counterexample. Connection to the applications.  More discussion. More experiments.
>
> We add a section in Appendix D in the revised manuscript to provide more discussions and results on the impossiblity theory. We briefly summarize it as follows:
>
> In our example in Section 4,  we consider binary features and label. This is inspired by the popular classification tasks in the literature of IRM, e.g., CMNIST, CifarMnist, ColoredObject, Waterbirds and CelebA. Take CMNIST for example. The label is 0 or 1. The invariant semantic feature, $X_1$, is the semantic feature  ‘0’ or ‘1’ of the digit shape. The spurious feature $X_2$ (color) is also binary: either red or green. So we also denote the binary invariant and spurious features as 0/1 in the following discussion.
>
> In Appendix D, we show that the construction of CMNIST is analogous to first data generation process  in our Example 1 in Section 4. Moreover, we construct a new dataset MCOLOR (short for MnistColor) that is analogous to the second data generation process in Example 1. In MCOLOR, the digit shape is the spurious feature. We show that CMNIST and MCOLOR have the same joint training distribution of color, digit shape and label. Our task is to learn  invariant features from the joint distribution. Due to EIIL’s inductive bias, it will deterministically rely on the digit shape as the invariant feature, resulting in poor performance in MCOLOR. However, since we have no prior knowledge of the invariant feature or the data generation process, the TRUE invariant feature could be color like the MCOLOR dataset.  LrF will also rely on either color or digit as the invariant feature, similar to EIIL. **Please see the detailed empirical results in Appendix D.** (We add the results of EIIL in Appendix D and we are still working on the experiments of LrF.)
>
>
> ### Q1-3. The SCM for the general impossibility theoretical result.
>
> Thanks for this comment. If we understand correctly (please correct us if we have any misunderstanding about ‘specific SCM’), in Theorem 1 we do not assume any specific form of SCMs, and each structural equation has a noisy (or exogenous) term and the causal mechanism can be either linear or non-linear. Specifically, Theorem 1 can work with the SCM of Equation (1) in our paper, and Equation (1) is already more general than Assumption 1 of [1] because $g_v(X_v, \epsilon_v)$ includes $g_v(X_v)+\epsilon_v$ and [1] uses linear function for $g_v(\cdot)$.
>
> [1] Invariance Principle Meets Information Bottleneck for Out-of-Distribution Generalization.

---

### Official Review · Reviewer_Ypmy · 2022-07-18

**Rating:** 7
**Confidence:** 4
**Soundness:** 3 good
**Presentation:** 3 good
**Contribution:** 3 good

**Summary:**

This paper theoretically proves that it's impossible to infer the environments purely from the heterogeneous data; consequently, additional information is definitely needed. It proposes a framework that jointly learns environment partitions and invariant representations with the additional information. Experimental results demonstrate the effectiveness of this method.

**Questions:**

1. What is the auxiliary information $Z$ in the experiment synthetic dataset shown in Section 7.1? There is no $Z$ in the synthetic process.

**Limitations:**

I haven't seen any obvious limitations except for the one mentioned in the last part of the paper.

**Strengths And Weaknesses:**

Strengths:

1. This paper theoretically proves that it's impossible to identify the invariant features purely from heterogeneous data, which is meaningful. The toy example is intuitive and easy to understand.

2. This paper proposes a framework to jointly learn environment partition and invariant representation, which is simple but equipped with a theoretical guarantee, which is also meaningful.

3. This paper is well-written and easy to follow.

Weaknesses:

1. I haven't seen the guarantee to demonstrate how informative the auxiliary information $Z$ should be to identify the invariant features. Looking through Assumptions 1-4 and Conditions 1-2, there seems to be no discussion on $Z$, which should be included in the paper.

---

> ### Author Response · Authors · 2022-08-02
> **Response**
>
> ## Weakness
>
> ### Q1 More discussion/demonstration on Z
>
> Thank you for the insightful question and we indeed miss this part in our original submission. In the revised version, we add a section (Appendix F) on the causal interpretations of $Z$ with specific causal graphs and examples. We will try to fit this part into the main paper in the future version because it is of vital importance.
>
> We briefly summarize the results in Appendix F.
>
> * **How to satisfy Condition 1**. The path between  $Z$ and $Y$ should be $d$-separated by $X_v$. Notably, $Z$ can’t be the parent or the child of $Y$ as shown in Figure 6. Figure 4 also shows a concrete example of the meta information in the image classification tasks together with a illustrating causal graph. The meta information of a image, e.g., the time slot, coordinate and temperate, does not have a direct effect on the target.  It has some correlations with some nodes of the causal system.  Then we have $H(Y|X_v)=H(Y|X_v, Z)$. This meta information serves as valid $Z$. Figure 5 illustrates more valid choices of $Z$ with causal explanations.
>
> *  **How informative are Z** (**How to satisfy Condition 2**). Condition 2 is hard to check in practice. While $Z$ cannot be the direct parent or the child of $Y$ (otherwise violating Condition 1), $Z$ should be correlated with some nodes in the interested causal graph. Further, we show an additional theoretical result in Appendix F.2: even if Condition 2 is only partially satisfied, we can still discard the spurious features that are distinguishable by the collected $Z$. At the same time, the invariant features are preserved. As we collect more $Z$ satisfying Condition 1, we can discard more spurious features.
>
> In conclusion, we should try to **find as many $Z$, which satisfies Condition 1 and is not independent of the nodes in the causal graph, as possible**.
>
> Please refer to Appendix F for the detailed contents.
> Thanks again for the constructive suggestions.
>
> ## Question
>
> ### Q1 The Z in the synthetic process in Section 7.1
>
> Thank you for pointing out the confusion in this part. We use the time index $t$ as the auxiliary information $Z$. We have added this in Section 7.1.

---

> > ### Comment · Reviewer_Ypmy · 2022-08-07
> > **Author Response Acknowledgment**
> >
> > Thank you for your clarifications! Then I have no further questions.

---

### Official Review · Reviewer_3T6p · 2022-07-18

**Rating:** 6
**Confidence:** 4
**Soundness:** 3 good
**Presentation:** 3 good
**Contribution:** 2 fair

**Summary:**

The authors propose an approach to learning an invariant representation, which simultaneously learns the representation along with an environment partition over samples based on auxiliary measurements, such as time indices or location metadata. They provide sufficient conditions for their proposed method to identify invariant features in both the feature selection and the linear feature learning setting. They also describe necessary conditions for the identifiability of invariant features in the feature selection setting. Finally, experiments demonstrate that the proposed method is effective at learning invariant features, almost matching the performance of IRM which has access to a ground-truth partition of the variables.

**Questions:**

**Major questions**
+ What is the intuition behind Assumption 3? In particular if we think of $X_v$ as the parents of $Y$ and $X_s$ as the children of $Y$?
+ I did not understand Corollary 1. First, I assume it is meant to be (a) _or_ (b) _or_ (c), not _and_. Second, what does “Index” denote? Why is $h$ supposed to be injective - can’t it map the same inputs to the same one-hot vector?

**Suggestions to help reader understanding**
+ It helps to describe new quantities directly after they are introduced. For example, after Equation (3), mention that $\rho$ softly partitions environments before introducing the other definitions, mention that $\mathcal{R}_{\rho^{(k)}}(\omega, \Phi)$ is the loss of $f_w$ in environment $k$, etc.
+ It may help to separately define the invariance penalty, including the max term. This emphasizes that the max term is only involved in this term, and directly highlights what is being proposed.
+ It would help to highlight the differences between Theorem 3 and the results for IRM, since they are so similar. It appears simply that you allow the environment partition to be learned from auxiliary information instead of being given, are there any other significant differences?

Minor points
+ Assumption 3 is unclear: you say adding another feature does not make the penalty diminish, but in the second sentence, it seems that it is allowed to become smaller, just that it is not allowed to vanish to zero.
+ After Equation (3), you use $w$ to subscript $f$ instead of $\omega$
+ Assumption 4: any “distinct” features, not “distant”


**Limitations:**

Limitations and potential negative impacts are adequately addressed.

**Strengths And Weaknesses:**

**Strengths**
+ The paper is well-written in most aspects: assumptions are cleanly stated and intuitively explained after they are introduced. The feature selection example provides an intuitive illustration of the ideas.
+ The paper is “significant” in an uncommon way. In particular, it seems that it will play an important regularizing role to correct the confusion present in some other works on invariant learning, which have sought to learn an environment partition without metadata. For instance, the authors correctly point out that EIIL can only be successful when the spurious features are *more* correlated with the label than the invariant features are. The presentation of necessary conditions in Propositions 1 and 2 is especially helpful for streamlining future literature on this topic.
+ The experiments are well-done, with the proposed method performing near the “oracle” baseline of IRM across a variety of different domains (synthetic, house price prediction, a CelebA experiment, and a land cover prediction task).

**Weaknesses**
+ Most of the results are not particularly surprising for those familiar with causality. For instance, Example 1 is a standard example of Markov equivalence.
+ Some of the results seem tautological, e.g., violating Condition 2 seems equivalent with the conclusion of Proposition 2 that a spurious feature will be included in the feature selection.

---

> ### Author Response · Authors · 2022-08-02
> **Response**
>
> ## Weakness
> ### Q1-1 The results are not particularly surprising for those familiar with causality, e.g., Example 1.
>
> We agree that the impossibility part in our paper is indeed the identifiability issue, a very fundamental and important concept. This issue is well-known to those who are familiar with causal inference or ICA, like the reviewer, and has been discussed in recent ML applications as well, e.g., [1][2]. However, it seems that for this new and practically meaningful setting (invariance learning without environment indexes, which has already attracted some attention in existing works) considered in our paper, the identifibility issue is omitted once again.
>
> We believe that a contribution of ours is to formulate this setting in the causal language and then adapt the causality techniques accordingly, as Reviewer 9XzW wrote “I’ve long thought that environment inference was asking too much so I really appreciate a clear counter example that shows where it can fail ... and the auxiliary information perspective is a nice way around the negative examples. ” So our result again points out an important issue and we do hope the identifibility issue can draw even more attention in the ML community.
>
> [1]  Challenging common assumptions in the unsupervised learning of disentangled representations. ICML  2018.
>
> [2] Variational autoencoders and nonlinear ICA: a unifying framework. AISTATS, 2020.
>
> ### Q1-2 Some of the results seem tautological.
>
> Thanks for the insightful comment. Violating Condition 2 indeed leads to the conclusion of Proposition 2 that some spurious features can be selected by the feature mask.  It may seem straightforward for one who is familiar with the proof of Theorem 2 to reach the conclusion of Proposition 2. We believe that adding Proposition 2 may help readers who are not familiar with the feature selection proof,  to see the necessity of Condition 2.
>
> We will try to make this part more concise and reduce the duplicated contents,  or put some of them into the appendix. Thanks again for pointing this out.
>
> ## Questions
>
> ### Q2-1 Major questions : - What is the intuition behind Assumption 3?
>
> This assumption aims to ensure the penalty not degenerate when $X_v$ is too informative of $Y$. Suppose $X_v$ is the parent of $Y$ and $X_s$ is the child of $Y$. If $X_v$ is not fully informative of $Y$, then $H(Y|X_v)>0$. So if  $H(Y|X_s, \rho(Z)) > H(Y|X_s)$, then we typically have $H(Y|X_s, X_v, \rho(Z)) > H(Y|X_s, X_v)$.  The following two cases are troublesome and are ruled out by Assumption 3:
> * If $X_v$ is fully informative of $Y$, i.e., $H(Y|X_v)=0$, then it will lead to $H(Y|Xs, Xv, \rho(Z)) = H(Y|X_s, X_v) = 0$. In this case, the penalty will be 0 no matter what the spurious feature $X_s$ is.
> * If $X_v$ is highly informative of $Y$, i.e., $H(Y|X_v)=10^{-6}$, then $H(Y|X_s,X_v,\rho(Z)) <= 10^{-6}$ and $H(Y|X_s,X_v)<=10^{-6}$. It leads to the penalty $H(Y|X_v, X_s) - H(Y|X_s,X_v,\rho(Z)) <= 10^{-6}$. Then the penalty is too small.
> Assumption 3 excludes the above cases where $X_v$ is too informative and the penalty vanishes.
>
> ### Q2-2 Major Questions: questions on Corollary 1 (Q2). ''I did not understand Corollary 1...same one-hot vector?''
>
> - Yes, it is '(a) or (b) or (c)'.
> - By 'Index', we mean that each sample has a distinct number associated with it. Taking a dataset with N data points for example. We can assign in $k=1,...,N$ to the samples.  Then $h(Index(X, y))$ assigns individual weights to each sample in the training dataset. We will add an explicit definition.
> - The injective condition just makes it easy for the proof. The injective requirement is not necessary. For example, the results still hold if $h(Y)$ (or $h(Index(X, Y))$) contains additional information of $Y$ conditional on $X_v$. Then we have $H(Y|X_v, \rho(Z)) = H(Y|X_v, h(Y)) < H(Y|X_v)$, which violates Condition 1.
>
> Thanks for this insightful comment to help improve our work.
>
> ### Q3-1 Suggestions (Q1-Q2): Describe new quantities, define the invariance penalty.
>
> Thanks a lot for the suggestions. We have modified this part accordingly in Section 5.1 in the revised manuscript, marked in blue.
>
> ### Q3-2 Suggestions: The relationship between our Theorem 3 and the results in IRM.
>
> We agree that our Theorem 3 is similar to the results of IRM. We provide Theorem 3 to illustrate that our framework is applicable to the linear feature learning setting. We will add more discussion on this result to enhance the readability and will consider to name it as a ‘Proposition’, as we do not have many technical contributions here.
>
> ### Q4-1 Minor Points. TYPOs.
>
> Thanks a lot for pointing out these typos. We have corrected them accordingly.

---

> > ### Comment · Reviewer_3T6p · 2022-08-05
> > **Author response acknowledgment**
> >
> > Thanks for clarifying the questions and taking the suggestions into account!

---

### Author Response · Authors · 2022-08-02
**General Response**

We thank all the reviewers for their time. We have uploaded a revised version of our paper, following the suggestions/comments from all the reviewers. Some major changes are:

- We add a detailed discussion on how to choose the auxiliary information in Appendix F, together with some causal interpretations of $Z$ and an additional theoretical result.
- We add more discussions on Example 1 in Appendix D. We also construct a dataset and provide additional empirical results to illustrate the impossibility example.
- We rephrase the methodology part in Section 5.1 for a better readability.
- We have also reported more experimental results, including GroupDRO as baselines, different choices of $K$ on CelebA data, visualizing the inferred environments in the CelebA experiment.

Thanks a lot for the effort from all the reviewers and program committee.

---

### Meta-Review · Area_Chair_4btG · 2022-08-26

**Recommendation:** Accept
**Confidence:** Certain

**Metareview:**

This paper has been well received by the reviewers - all reviewers are positive including significant revisions upwards after rebuttal. Notable strengths are clarifying when you can/cannot identify environments for invariant learning and proposing sufficient and necessary conditions for the same. Further some reviewers have expressed positive opinion on the experiments of the paper which  is valuable as well.

To the authors: Please do take into account reviewers questions when preparing camera ready.



**Award:**

No

---

### Decision · Program_Chairs · 2022-09-14

Accept